# Building the vertebrate codex using the gene breaking protein trap library

Noriko Ichino[1], MaKayla R Serres[1], Rhianna M Urban[1], Mark D Urban[1], Anthony J Treichel[1], Kyle J Schaefbauer[1], Lauren E Tallant[1], Gaurav K Varshney[2,3], Kimberly J Skuster[1], Melissa S McNulty[1], Camden L Daby[1], Ying Wang[4], Hsin-kai Liao[4], Suzan El-Rass[5], Yonghe Ding[1,6], Weibin Liu[1,6], Jennifer L Anderson[7], Mark D Wishman[1], Ankit Sabharwal[1], Lisa A Schimmenti[1,8,9], Sridhar Sivasubbu[10], Darius Balciunas[11], Matthias Hammerschmidt[12], Steven Arthur Farber[7], Xiao-Yan Wen[5], Xiaolei Xu[1,6], Maura McGrail[4], Jeffrey J Essner[4], Shawn M Burgess[2], Karl J Clark[1]*, Stephen C Ekker[1]*

[1]Department of Biochemistry and Molecular Biology, Mayo Clinic, Rochester, United States; [2]Translational and Functional Genomics Branch, National Human Genome Research Institute, National Institutes of Health, Bethesda, United States; [3]Functional & Chemical Genomics Program, Oklahoma Medical Research Foundation, Oklahoma City, United States; [4]Department of Genetics, Development and Cell Biology, Iowa State University, Ames, United States; [5]Zebrafish Centre for Advanced Drug Discovery & Keenan Research Centre for Biomedical Science, Li Ka Shing Knowledge Institute, St. Michael's Hospital, Unity Health Toronto & University of Toronto, Toronto, Canada; [6]Department of Cardiovascular Medicine, Mayo Clinic, Rochester, United States; [7]Department of Embryology, Carnegie Institution for Science, Baltimore, United States; [8]Department of Clinical Genomics, Mayo Clinic, Rochester, United States; [9]Department of Otorhinolaryngology, Mayo Clinic, Rochester, United States; [10]Genomics and Molecular Medicine Unit, CSIR– Institute of Genomics and Integrative Biology, Delhi, India; [11]Department of Biology, Temple University, Philadelphia, United States; [12]Institute of Zoology, Developmental Biology Unit, University of Cologne, Cologne, Germany

*For correspondence:
Clark.Karl@mayo.edu (KJC);
ekker.stephen@mayo.edu (SCE)

**Competing interests:** The authors declare that no competing interests exist.

**Abstract** One key bottleneck in understanding the human genome is the relative under-characterization of 90% of protein coding regions. We report a collection of 1200 transgenic zebrafish strains made with the gene-break transposon (GBT) protein trap to simultaneously report and reversibly knockdown the tagged genes. Protein trap-associated mRFP expression shows previously undocumented expression of 35% and 90% of cloned genes at 2 and 4 days post-fertilization, respectively. Further, investigated alleles regularly show 99% gene-specific mRNA knockdown. Homozygous GBT animals in *ryr1b*, *fras1*, *tnnt2a*, *edar* and *hmcn1* phenocopied established mutants. 204 cloned lines trapped diverse proteins, including 64 orthologs of human disease-associated genes with 40 as potential new disease models. Severely reduced skeletal muscle $Ca^{2+}$ transients in GBT *ryr1b* homozygous animals validated the ability to explore molecular mechanisms of genetic diseases. This GBT system facilitates novel functional genome annotation towards understanding cellular and molecular underpinnings of vertebrate biology and human disease.

**eLife digest** The human genome counts over 20,000 genes, which can be turned on and off to create the proteins required for most of life processes. Once produced, proteins need move to specific locations in the cell, where they are able to perform their jobs. Despite striking scientific advances, 90% of human genes are still under-studied; where the proteins they code for go, and what they do remains unknown.

Zebrafish share many genes with humans, but they are much easier to manipulate genetically. Here, Ichino et al. used various methods in zebrafish to create a detailed 'catalogue' of previously poorly understood genes, focusing on where the proteins they coded for ended up and the biological processes they were involved with.

First, a genetic tool called gene-breaking transposons (GBTs) was used to create over 1,200 strains of genetically altered fish in which a specific protein was both tagged with a luminescent marker and unable to perform its role. Further analysis of 204 of these strains revealed new insight into the role of each protein, with many having unexpected roles and localisations. For example, in one zebrafish strain, the affected gene was similar to a human gene which, when inactivated, causes severe muscle weakness. These fish swam abnormally slowly and also had muscle problems, suggesting that the GBT fish strains could 'model' the human disease.

This work sheds new light on the role of many previously poorly understood genes. In the future, similar collections of GBT fish strains could help researchers to study both normal human biology and disease. They could especially be useful in cases where the genes responsible for certain conditions are still difficult to identify.

## Introduction

Analyses of genomic sequences from over 100 vertebrate species (*Meadows and Lindblad-Toh, 2017*) have revealed that we need more than nucleic acid sequence alone to comprehend the vertebrate genome. A more complete understanding of any genetic locus requires knowledge of its expression pattern and its function(s) in subcellular, cellular, and organismal contexts—the compendium of information that can be described as a gene 'codex'. Despite their importance, the expression patterns and functions of most protein coding genes remain surprisingly uncharacterized. The number of these genes linked to human disease without functional insights into their gene-disease relationships highlights the significance of this knowledge gap (*Kettleborough et al., 2013*). In recent estimates, 80% of rare, undiagnosed diseases are thought to have genetic underpinnings (*Robe, 2005*; *Varga et al., 2018*; *Wangler et al., 2017*). Tools are therefore needed to identify and annotate the expression and function(s) of these poorly characterized gene products in both biological and pathological processes.

Zebrafish (*Danio rerio*) has emerged as an outstanding model to bridge the gap between sequence and function in the vertebrate genome. Investigations of gene function in zebrafish, from organismal to subcellular, are amenable to both forward and reverse genetic approaches (*Stoeger et al., 2018*). Additionally, the natural transparency of developing zebrafish enables live, non-invasive collection of gene expression data at a subcellular resolution on an organismal scale. Therefore, the zebrafish facilitates parallel discovery of gene expression and function towards a comprehensive codex of the vertebrate genome. To begin constructing this vertebrate codex, we previously developed a unique, revertible mutagenesis tool called the gene-break transposon (GBT) with elements that cooperate to report gene sequence, expression pattern, and function (*Clark et al., 2011a*). Specifically, when integrated in the sense orientation of a transcriptional unit, the GBT protein trap overrides endogenous splicing and creates a fusion between upstream exons and its start-codon deficient monomeric RFP (mRFP) reporter. Then, the strong internal polyadenylation site and putative border element following the mRFP truncate the gene product. Finally, the GBT construct is flanked by loxP sites on either side to enable excision and subsequent rescue with Cre-recombinase (*Clark et al., 2011a*).

Visualization of the start-codon deficient mRFP reporter requires an in-frame integration. In the original GBT protein trap construct, RP2.1 (*Figure 1A*), this in-frame requirement restricts mRFP expression to a single reading frame and leaves the potential to truncate genes without reporting

their expression with mRFP (*Clark et al., 2011a*). We therefore developed a new series of GBT protein trap constructs, including versions to trap expression in each of the three potential reading frames (*Figure 1A*). Alongside the original, we employed these new vectors in zebrafish to generate and catalog over 800 additional GBT protein trap lines with visible mRFP expression at 2 days post-fertilization (dpf) (end of embryogenesis) or four dpf (larval stage). 147 of these additional GBT lines were cloned, and candidate genes were identified for another 144 GBT lines. mRFP expression in

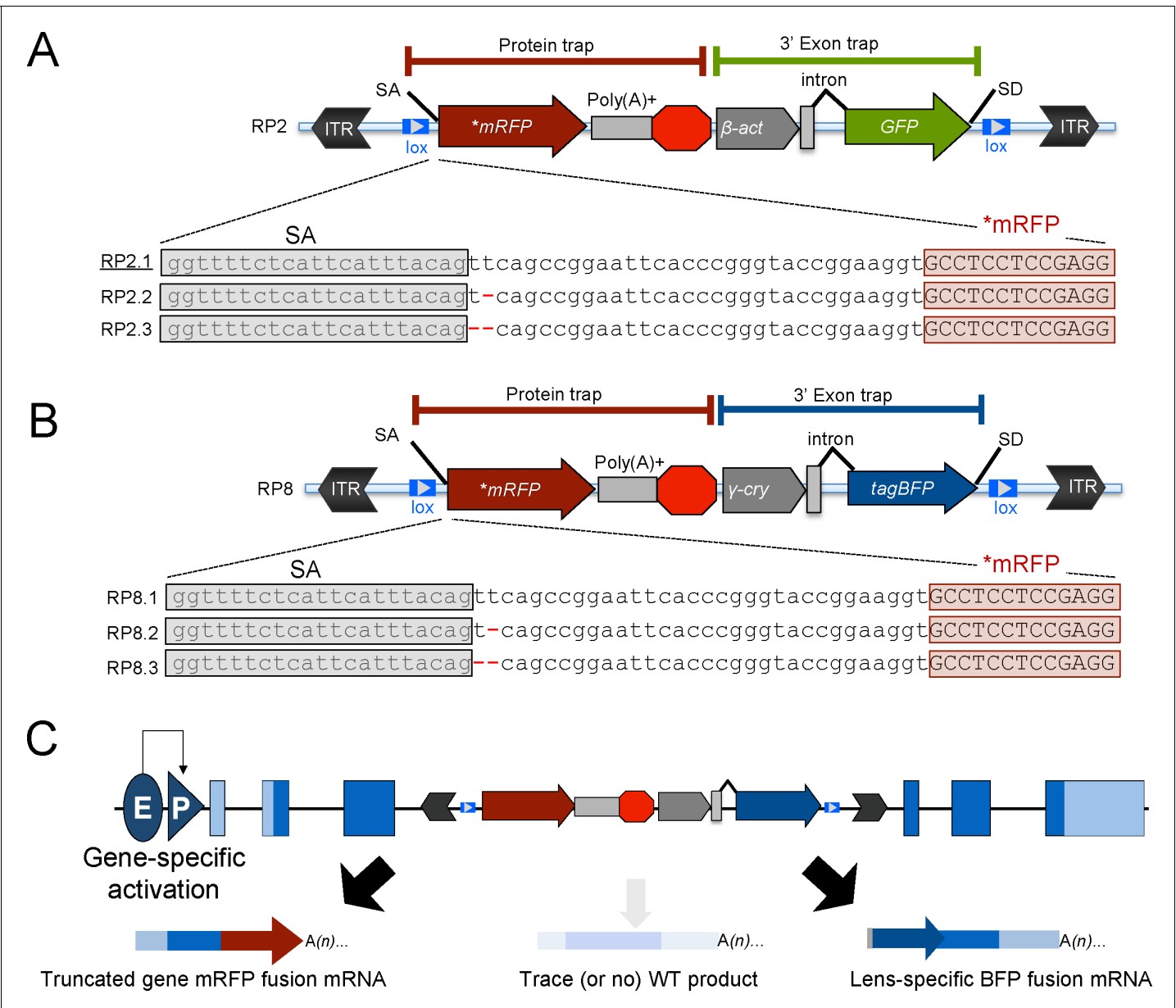

**Figure 1.** Schematic of the RP2 and RP8 gene-break transposon (GBT) system with all three reading frames of AUG-less mRFP reporter. (**A–C**) Schematic of the GBT system, RP2 and RP8 incorporate a protein-trap cassette fused with three reading frames of AUG-less mRFP reporter and a 3' exon trap cassette with GFP or tagBFP reporters, respectively. (**A**) RP2 series (RP2.1, RP2.2 and RP2.3). Underline: Previously published vector construct (**B–C**) RP8 series (RP8.1, RP8.2 and RP8.3) with a schematic RP8 insertion event showing expected transcription off of a locus below (**C**). ITR: inverted terminal repeat, SA: splice acceptor, lox: Cre recombinase recognition sequence, *mRFP: AUG-less mRFP sequence, poly (A)+: polyadenylation signal, red octagon: extra transcriptional terminator and putative border element, *β-act*: carp beta-actin enhancer, *γ-cry*: gamma crystalline promoter, SD: splice donor, E: enhancer, P: promoter, and WT: wild-type.

The online version of this article includes the following figure supplement(s) for figure 1:

**Figure supplement 1.** Representative expression patterns of mRFP fusion protein integrated all reading frames of RP2 and RP8.

cloned GBT lines showcased novel expression patterns for a population of genes encoding diverse proteins in function and localization, including 64 implicated in human disease. Further, animals homozygous for the GBT allele in *ryr1b* displayed severely dampened skeletal muscle Ca$^{2+}$ transients, demonstrating the ability to elucidate molecular mechanisms of genetic disorders. Since detailed investigations of mutant phenotypes are vital to functional annotation of the vertebrate genome, the mutagenic reporters in our GBT system provide the basis for this functional annotation to better understand normal biology and human disease.

## Results

### GBT vector series RP2 and RP8 illuminate all three vertebrate proteomic reading frames

We previously reported the intron-based gene-break transposon (GBT) as an effective and revertible loss-of-function tool for zebrafish (*Clark et al., 2011a*). The original GBT construct called RP2/RP2.1 contains the following key features (*Figure 1A*): 1) flanking miniTol2 sequences for transposase-mediated random integration (*Balciunas et al., 2006*; *Kawakami et al., 2004*; *Urasaki et al., 2006*), 2) a 5′ protein trap containing a strong splice-acceptor (SA) and a start codon-free mRFP reporter to detect 5′ sequence and visualize in vivo expression of the trapped locus with the endogenous promoter (*Clark et al., 2011a*; *Ding et al., 2013*; *Ding et al., 2017*; *Liao et al., 2012*; *Petzold et al., 2009*; *Westcot et al., 2015*; *Xu et al., 2012*), 3) a mutagenic transcriptional terminator containing both a polyadenylation signal (pA) and a putative border element to truncate the trapped locus in conjunction with the protein trap (*Sivasubbu et al., 2006*), 4) a 3′ exon trap with a β-actin promoter driving expression of GFP to report 3′ sequence and detect lines with weak (or absent) mRFP expression or with the integration in other frames of the mRFP reporter. (*Clark et al., 2011a*; *Petzold et al., 2009*; *Sivasubbu et al., 2006*), 5) a second mini-intron within the GFP expression cassette that can further contribute to loss of wild-type transcripts, and 6) flanking loxP sites for Cre-mediated excision and restoration of trapped locus function using both germline (*Petzold et al., 2009*) and somatic approaches (*Clark et al., 2011a*; *Ding et al., 2013*; *Westcot et al., 2015*).

Initial experiments with the RP2.1 construct, however, revealed some limitations. First, effective transcript trapping does not always generate mRFP reporter expression because the RP2.1 plasmid is designed for a single reading frame. Molecular cloning of GFP$^{+}$/mRFP$^{-}$ lines demonstrated the requirement to capture an appropriate reading frame to visualize the mRFP reporter. Even though RP2.1 is designed to use one main reading frame, some lines with mRFP expression used an alternate 'CAG' five nucleotides downstream of the main splice acceptor which offered a second chance at creating a functional mRFP reporter (*Clark et al., 2011a*). Therefore, to maximize genome coverage of our mutagenesis vectors in this study, we created a series of RP2 constructs to encode functional mRFP in each of the three reading frames (*Figure 1A*). Second, the 3′ exon trap in the RP2 series uses the nearly ubiquitous β-actin promoter to drive expression of GFP (detectable around the seven- to eight-somite-stage similar to ubiquitous GFP expression driven under the *EF1α* enhancer/promoter [*Davidson et al., 2003*]) which could interfere with another GFP-based reporter system in future studies (*Clark et al., 2011a*). To overcome this limitation, we engineered a novel, next-generation GBT series called RP8. RP8 constructs possess a new 3′ exon trap cassette that uses the γ-crystalline promoter to drive expression of lens-specific tagBFP instead of the ubiquitous expression GFP with RP2 series vectors. (*Figure 1B–C*). Additionally, all RP8 series for three reading frames reporting mRFP constructs are built on a smaller vector backbone and include new restriction enzyme sites that render these vectors modular for subsequent genetic engineering. Using all five of these new GBT constructs in zebrafish, we conducted an initial screen for expression of protein trap mRFP and observed that all RP2 and RP8 series constructs readily produced mRFP fusion proteins expressed from their endogenous promoters (*Figure 1—figure supplement 1*).

### Creation of a GBT-line collection enables illumination of the vertebrate genome

We then deployed all of these GBT vectors to generate over 800 additional zebrafish GBT lines. A key feature of the protein trap in these lines is the ability to non-invasively image the spatial and temporal expression patterns of the trapped loci. Our initial mRFP$^{+}$ lines demonstrated that, using

standard methods, this imaging was going to be a major bottleneck (*Figure 2A*). Consequently, we utilized both SCORE imaging—a capillary tube placed in a refractive index-matched medium for efficient sample rotation—on an ApoTome (*Petzold et al., 2010*, and see Materials and methods) or a Zeiss Lightsheet Z.1 SPIM microscope (see Materials and methods) to enable high throughput fluorescence imaging. To date, we have now cataloged over 1,200 GBT lines with robust mRFP expression in heterozygous F2 animals at two dpf and/or four dpf according to our screening pipeline and made all imaging data freely accessible on zfishbook (www.zfishbook.org) (*Clark et al., 2012*; *Figure 2A*). We have cryopreserved these 1,200 GBT lines and have retained them at the Mayo Clinic Zebrafish Facility (MCZF) with a copy also sent to the Zebrafish International Resource Center (ZIRC) (*Figure 2A*).

## Molecular cloning of a subset of GBT lines highlights the genetic diversity of this protein trap collection

Traditional molecular methods, such as inverse PCR (iPCR) and thermal asymmetric interlaced (TAIL) PCR and 5' and 3' rapid amplification of cDNA ends (RACE), are labor-intensive and represent a functional bottleneck in identifying randomly integrated loci in GBT lines. To overcome this, we employed a rapid cloning process based on methods used to isolate retroviral integrations that leverage the massive parallel sequencing technology of the Illumina MiSeq and a custom bioinformatics pipeline that involves both mapping and annotation (*Figure 2B*; *Varshney et al., 2013a*; *Varshney et al., 2013b*). First, fin-clips from four male animals per GBT line were obtained during sperm cryopreservation and used as a source of DNA for cloning the integrated locus. Next, high-throughput sequencing amplified reads with barcodes linked to the source of DNA from the sperm-cryopreserved males. Mapping reads to the genome indicated potential GBT integration loci in each individual. A shared integration locus in multiple individuals from a single GBT line was considered a candidate integrated locus, and we termed a GBT line with at least one candidate integrated locus a 'GBT-candidate line'. After a candidate was determined using the sequencing pipeline or other manual molecular approaches, such as 5' RACE, 3' RACE, iPCR, or TAIL PCR, we used standard PCR to test if the candidate integration locus segregates with mRFP expression.

At the end of this pipeline, 204 GBT-candidate lines met the highest stringency of confirmed expression linkage and were classified as 'GBT-confirmed lines' (*Figure 2A*). While 57 of these GBT-confirmed lines have been previously published (*Clark et al., 2011a*; *Ding et al., 2013*; *Ding et al., 2017*; *El-Rass et al., 2017*; *Ma et al., 2020*; *Westcot et al., 2015*), 147 of these GBT-confirmed lines are newly characterized in this manuscript and were selected for confirmation based upon their expression pattern and/or homozygous phenotype (*Supplementary file 1*). A small subset of these GBT-confirmed lines mapped to areas in the genome without annotated transcripts. Publicly available RNA-sequencing data (*White et al., 2017*) revealed reads flanking a majority of these mapped integrations. Some of these reads contained evidence of splicing in the sense orientation of the mRFP reporter (*Supplementary file 1*). Finally, another 144 GBT-candidate lines from this pipeline have yet to be confirmed (*Figure 2A*). Integration locus annotation of both GBT-candidate and GBT-confirmed lines is available on zfishbook (www.zfishbook.org) (*Clark et al., 2012*).

## RP2.1 induces high knockdown efficiency of endogenous transcripts in GBT-confirmed lines

We wanted to determine the knockdown efficacy of the GBT system as a quantitative assessment of mutagenicity. We therefore compiled qRT-PCR data to compare wild-type and truncated, mRFP-fused transcript levels for all 26 RP2.1-derived GBT-confirmed lines that we and others have tested (*Clark et al., 2011a*; *Ding et al., 2013*; *Ding et al., 2017*, GBT0235—this manuscript). This compilation determined at minimum 97% knockdown in animals homozygous for the RP2.1 alleles. We next directly compared the transcriptional effects of RP2.1 with those of other published transposon-based protein trap systems (*Figure 3*). The FlipTrap system produced a range of 70–96% knockdown in six tested fish alleles (*Trinh et al., 2011*), similar to our initial R-series protein trap vectors (R14-R15) that contained a single splice acceptor and a simple transcriptional terminator (*Liao et al., 2012*; *Petzold et al., 2009*). The pFT1, which contains a single splice acceptor, but a tandem array of five simple polyadenylation sites, appears to be an improvement over these systems with 89–94% knockdown from four tested fish alleles (*Ni et al., 2012*). The 97% minimum knockdown observed

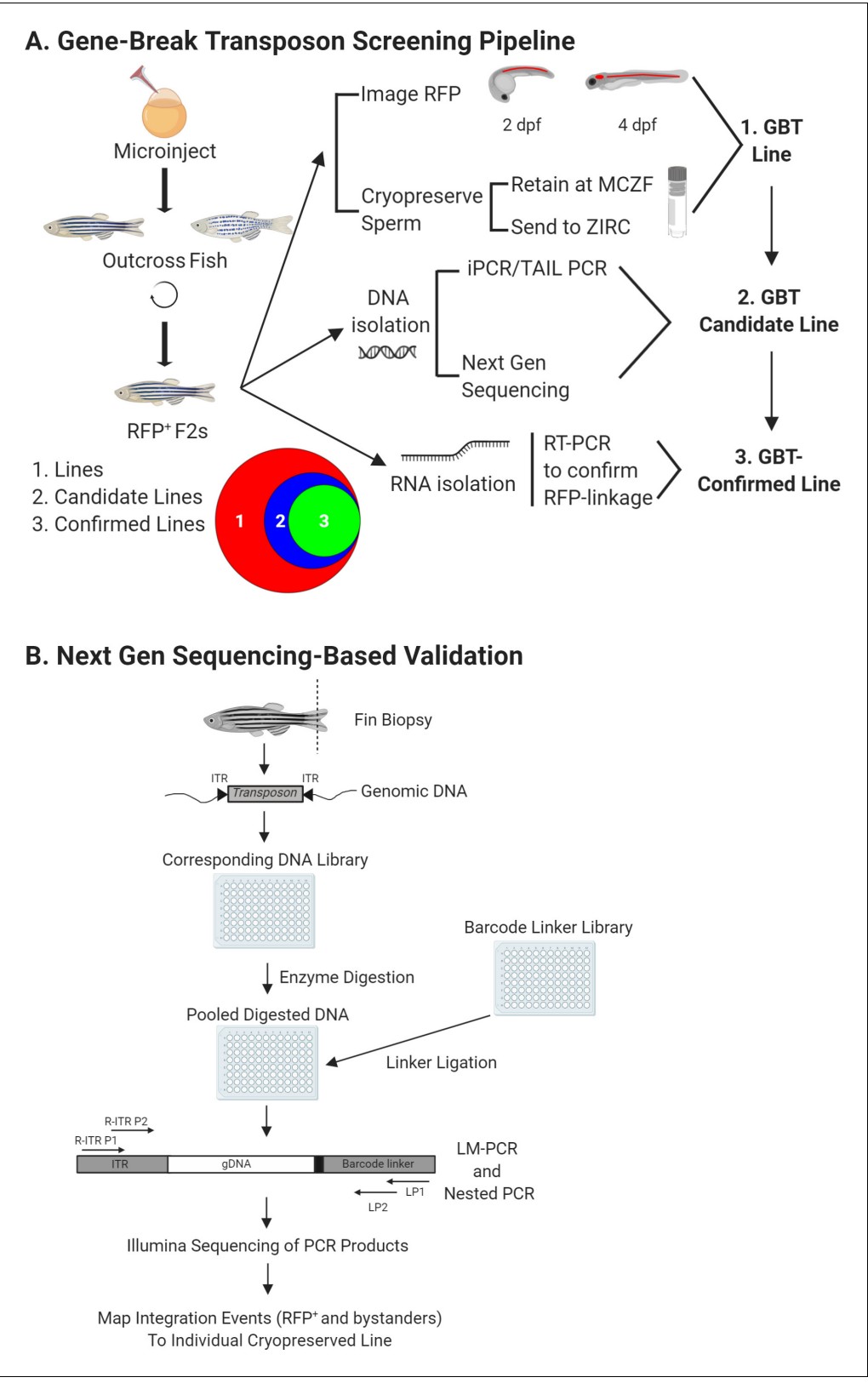

**Figure 2.** GBT screening pipeline. (**A**) Overview of GBT screening pipeline. Wild-type embryos at 1 cell were co-injected with RP plasmid and Tol2 transposase mRNA to create F0 founders. These F0 larvae were screened for non-mosaic RP expression, raised, and outcrossed for two generations. Then, mRFP+ F2 heterozygous larvae were 3-dimensionally imaged at 2 and 4 dpf and this imaging data were uploaded to zfishbook (http://www.zfishbook.

*Figure 2 continued*

org/). Sperm from four F2 males in over 1200 robust mRFP expressing lines were cryopreserved using the Zebrafish International Resource Center (ZIRC) standard protocol and stored at both ZIRC and Mayo Clinic Zebrafish Core Facility (MCZF). DNA and RNA isolated from these four F2 males with cryopreserved sperm was utilized to perform next-generation sequencing and to confirm RFP linkage of candidate lines by manual PCRs (iPCR, TAIL-PCR, 5' RACE and 3' RACE). Venn diagram illustrates current library of over 1,200 GBT lines with 204 GBT-confirmed lines out of 348 molecularly analyzed GBT-candidate lines. (B) Next generation sequencing based validation for GBT integration loci. Fin biopsies from four F2 males were utilized as DNA source for the validation process to identify GBT integration loci. Extracted genomic DNA was fragmented, pooled in 96-wells plate, and ligated with barcode linker to identify each single male with cryopreserved sperm. Linker-mediated (LM) PCR with the primers, R-ITR P1 and LP1 and nested PCR with the primers, R-ITR P2 and LP2 were conducted to perform Illumina sequencing the final PCR products. The integration events of individual sperm-cryopreserved male were mapped on zebrafish reference genome sequence with bioinformatics analysis. This figure was created with BioRender.com. The area proportional Venn diagram was produced using BioVenn (http://www.biovenn.nl/).

with RP2.1 is quantitatively higher than these other systems and could be deployed with other insertional genome engineering approaches. We expect the RP8 series vectors to have similar transcriptional knockdown to RP2.1, but to date we have not quantified the effect to measure their mutagenicity.

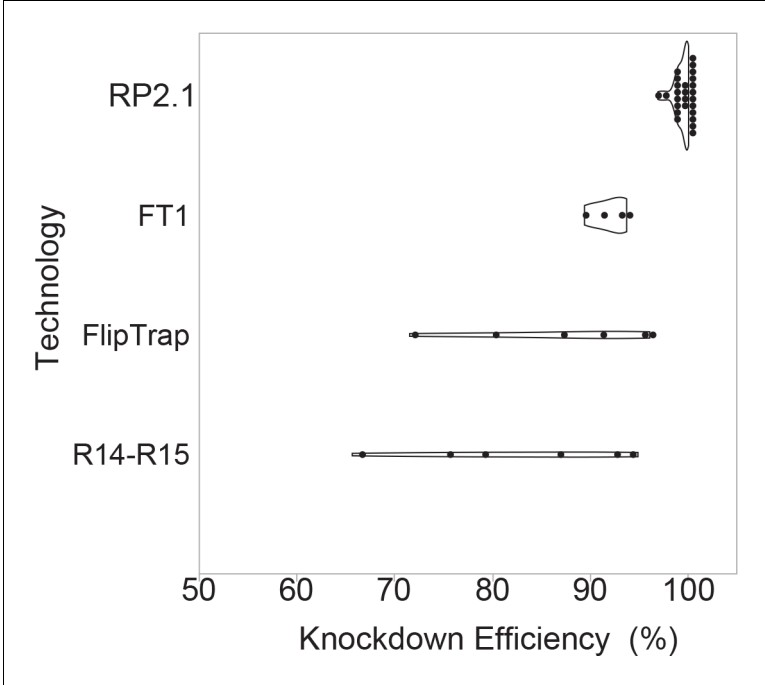

**Figure 3.** Knockdown efficiency of RP2.1 compared with previous gene-trap systems. Violin plots comparing percent knockdown efficiency in the analyzed individual lines generated by four protein trap systems. All plots show median. The data of previous protein trap systems were converted from the data in the original articles, R14-R15, our initial R-series protein trap vectors (n = 6), (*Clark et al., 2011a*; FlipTrap, FlipTrap vectors (n = 6), *Trinh et al., 2011*; FT1, FT1 vector (n = 4), *Ni et al., 2012*; RP2.1, RP2.1 vector (n = 26), *Clark et al., 2011a*; *Ding et al., 2013*; *Ding et al., 2017*; *El-Rass et al., 2017*; *Westcot et al., 2015* and unpublished data) (*Figure 3—source data 1*). The graph was made in JMP14 (SAS, Cary, NC).
The online version of this article includes the following source data for figure 3:

**Source data 1.** Numeric data analyzing knockdown efficiency in *lrpprc*$^{mn0235Gt/mn0235Gt}$.

## Phenotype appearance rate in GBT lines is comparable to other mutagenic technologies for forward genetic screening

In parallel to knockdown efficacy, we wanted to know the GBT construct effectiveness at mutagenizing its integrated locus. To assess this mutagenic efficiency, we conducted an initial phenotypic screen on F2 embryos and early larvae. In 179 mRFP$^+$ GBT lines, we identified 12 recessive phenotypes visible during the first five days of development that, among others, included lethal, cardiac, muscular, and integumentary defects as reported in previous studies (*Supplementary file 2*).

Molecular analyses revealed that a subset of these phenotypes stem from GBT integrations in genes with established loss of function mutant phenotypes including *ryr1b, tnnt2a, fras1, hmcn1*, and *edar* (*Clark et al., 2011a*; *Westcot et al., 2015*; Hatzold J et al., unpublished). In accord with their mRNA knockdown potency, these five GBT-confirmed lines (*ryr1b, tnnt2a, fras1, hmcn1*, and *edar*) phenocopied their respective homozygous mutants generated with other strategies and thereby validated the mutagenicity of the GBT constructs. To date, 17 of our GBT-confirmed lines have been published with homozygous phenotypes ranging from embryonic lethal, to reduced adult viability, to differences in pharmacological susceptibility (*Supplementary file 2*). Additional GBT-confirmed lines with homozygous phenotypes will continue to be identified and characterized in future studies.

## *ryr1b* confirmed line is a pioneer model of human disease that validates GBT ability to functionally annotate a genetic locus

We next sought to validate the ability of GBT constructs to functionally annotate genes. During the initial GBT library creation, we identified a GBT line with skeletal muscle-specific mRFP expression and a slow swimming phenotype (*Clark et al., 2011a*). We confirmed that this line possesses an RP2.1 integration between exon 81 and exon 82 of *ryr1b* and designated it *ryr1b$^{mn0348Gt}$*. Animals homozygous for this *ryr1b$^{mn0348Gt}$* allele show 97% knockdown of wild-type *ryr1b* mRNA levels (*Clark et al., 2011a*). Due to the well-characterized nature of *ryr1b* from the *relatively relaxed* mutant (*Hirata et al., 2007*), this line was ideal for a proof of concept experiment in this study to validate functional genome annotation with GBT constructs. *ryr1b* orthologs are known across species to encode calcium-activated calcium channels that release sarcoplasmic reticulum Ca$^{2+}$ stores to facilitate excitation-contraction coupling in skeletal muscles (*Hernández-Ochoa et al., 2015*; *Hirata et al., 2007*). Therefore, we set out to test whether loss of ryr1b in *ryr1b$^{mn0348Gt/mn0348Gt}$* animals dampens skeletal muscle Ca$^{2+}$ transients and may explain their previously reported slow swimming phenotype (*Clark et al., 2011a*).

To address this, we injected the skeletal muscle-targeted construct *p-mylpfa:GCaMP3* (*Baxendale et al., 2012*) into both *ryr1b$^{+/+}$* and *ryr1b$^{mn0348Gt/mn0348Gt}$* animals, treated these animals at 2 dpf with 20 mM pentylenetetrazole (PTZ) to maximize the probability of recording muscle activity, and assayed individual skeletal muscle Ca$^{2+}$ transients associated with PTZ-induced convulsions (*Figure 4A*). *p-mylpfa:GCaMP3* injection at the single-cell stage resulted in mosaic-labeled, GCaMP3$^+$ myocytes in both *ryr1b$^{+/+}$* and *ryr1b$^{mn0348Gt/mn0348Gt}$* animals (*Figure 4B,F*). Likewise, PTZ-treated *ryr1b$^{+/+}$* and *ryr1b$^{mn0348Gt/mn0348Gt}$* animals showed spontaneous, convulsion-associated Ca$^{2+}$ transients in their myocytes at two dpf (*Figures 4C–E,G–I*). However, PTZ-induced Ca$^{2+}$ transients in myocytes of *ryr1b$^{+/+}$* animals had higher peak amplitude when averaged within fish (*Figure 4J–K*) or within myocytes (*Figure 4—figure supplement 1A*) and shorter rise time (*Figure 4J,M*) than those in *ryr1b$^{mn0348Gt/mn0348Gt}$* animals. PTZ-induced Ca$^{2+}$ transient peak-width (*Figure 4J,L*) and decay time (*Figure 4J,N*) were not significantly different between *ryr1b$^{+/+}$* and *ryr1b$^{mn0348Gt/mn0348Gt}$* animals. In contrast, myocytes in *ryr1b$^{+/+}$* animals had more Ca$^{2+}$ transients during the imaging period than myocytes in *ryr1b$^{RP2.1/RP2.1}$* animals (*Figure 4—figure supplement 1B*). We were thus able to use a GBT-confirmed line to demonstrate that a smaller peak amplitude (consistent with *relatively relaxed* [*Hirata et al., 2007*]), slower upstroke, and lower frequency of Ca$^{2+}$ transients in skeletal muscle likely provide the basis for the slow swimming phenotype in *ryr1b$^{mn0348Gt/mn0348Gt}$* animals. The consistency of our findings in *ryr1b$^{mn0348Gt/mn0348Gt}$* with those in *relatively relaxed* mutants (*Hirata et al., 2007*) validates the functional genome annotation available with the GBT mutagenic system.

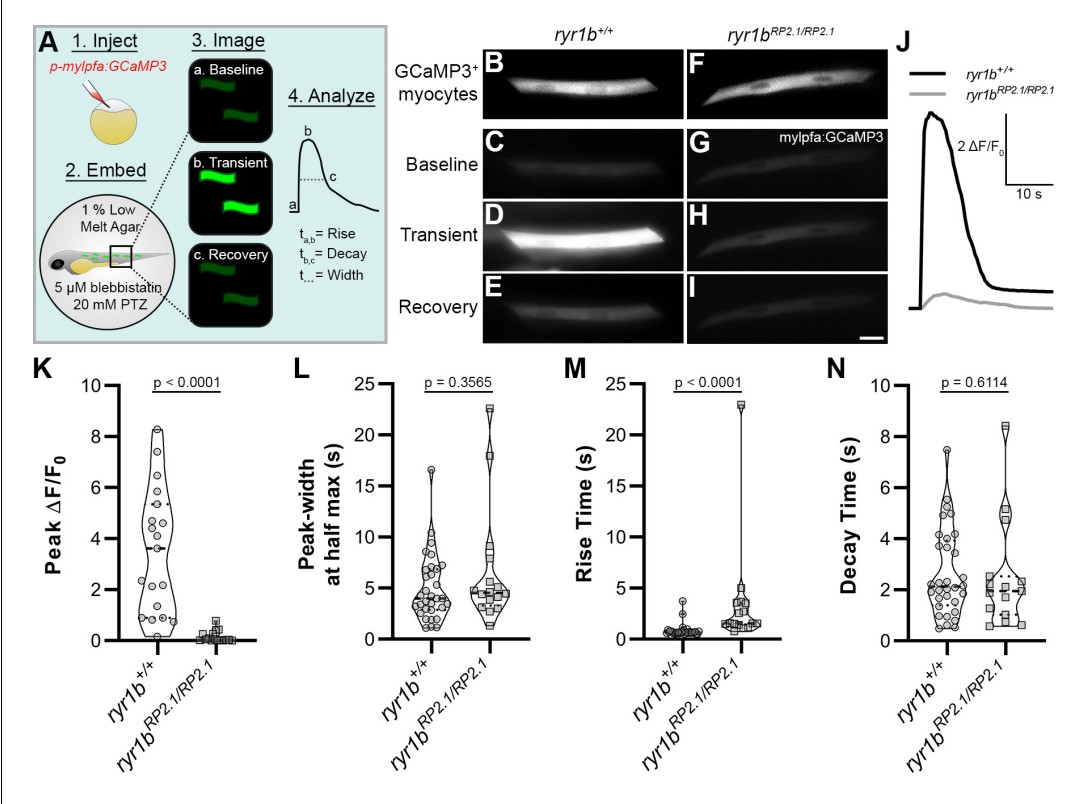

**Figure 4.** GBT demonstrates that neural disinhibition mediated $Ca^{2+}$ transients in $mylpfa^+$ myocytes require the ryanodine receptor *ryr1b* in vivo. (**A**) Cartoon showing approach to assay $Ca^{2+}$ transients in zebrafish myocytes through (1) injection of *p-mylpfa:GCaMP3* (*Baxendale et al., 2012*) at the single cell stage, (2) embedding in 1% low melt agar/20 mM pentylenetetrazole (PTZ)/5 μM (S)-(-)-blebbistatin, (3) imaging for 3 min to record transient-associated changes in myocyte GCaMP3 fluorescence at 2 days post-fertilization, and (4) $Ca^{2+}$ transient analysis. (**B–I**) Static images of GCaMP3 expressing myocytes (**B, F**) and representative GCaMP3 time-series images showing baseline (**C, G**), transient peak (**D, H**), and recovery (**E, I**) in $ryr1b^{+/+}$ (**C–E**) and $ryr1b^{mn0348Gt/mn0348Gt}$ (**G–I**) animals, respectively. Scale bar = 20 μm. (**J**) Representative $\Delta F/F_0$ traces of $Ca^{2+}$ transients from $ryr1b^{+/+}$ (black) and $ryr1b^{mn0348Gt/mn0348Gt}$ (gray) myocytes. (**K–N**) Violin plots comparing transient peak $\Delta F/F_0$ (averaged within fish) (**K**), $Ca^{2+}$ transient peak-width (**L**), $Ca^{2+}$ transient rise (**M**) and decay (**N**) time between $ryr1b^{+/+}$ and $ryr1b^{mn0348Gt/mn0348Gt}$ animals. All plots show median with interquartile range. For (**K**) $n_{ryr1b+/+}$ = 19 animals, $n_{ryr1bmn0348Gt/mn0348Gt}$ = 16 animals. For (**L–M**) $n_{ryr1b+/+}$ = 32 cells, $n_{ryr1bmn0348Gt/mn0348Gt}$ = 16 cells. For (**N**) $n_{ryr1b+/+}$ = 32 cells, $n_{ryr1bmn0348Gt/mn0348Gt}$ = 15 cells. Data are compiled from four independent experiments containing at least two animals in each group. p-values determined using the Mann-Whitney U test. Effect size (Cohen's d)=1.829 (**K**) and 0.866 (**M**). Source data can be found in *Figure 4—source data 1* (**K, L, M, N**) and *Figure 4—source data 2* (**J**).

The online version of this article includes the following source data and figure supplement(s) for figure 4:

**Source data 1.** Summary data analyzing the parameters of $Ca^{2+}$ transients in individual tested animals.
**Source data 2.** Individual $\Delta F/F_0$ traces of GCaMP3-fluorescence in both $ryr1b^{+/+}$ and $ryr1b^{mn0348Gt/mn0348Gt}$ myocytes.
**Figure supplement 1.** $Ca^{2+}$ transients in $ryr1b^{+/+}$ myocytes have higher peak amplitude and are more frequent than in $ryr1b^{mn0348Gt/mn0348Gt}$ myocytes.
**Figure supplement 1—source data 1.** Summary data analyzing the parameters of $Ca^{2+}$ transients in individual tested cells.

## GBT protein trapping generates a variety of potential models of human disease

This functional genome annotation available with the GBT system in zebrafish is powerful for understanding the genetic causes of human disease. For instance, mutations in *RYR1*, the human ortholog of *ryr1b*, are well-associated with a rare genetic neuromuscular disorder called central core disease. Central core disease commonly presents with mild to severe muscle weakness (*Jungbluth et al., 2018*) which is analogous to the slow swimming phenotype we saw in our $ryr1b^{mn0348Gt/mn0348Gt}$ animals (*Clark et al., 2011a*) and likely arises from similar disruptions to skeletal muscle $Ca^{2+}$ transients (*Figure 4*). A subset of this GBT collection consequently represents a potential library of human disease models. Intriguingly, 82% of OMIM listed human disease-associated genes (2601 genes) can be related to at least one zebrafish ortholog (*Howe et al., 2013*).

We therefore took a new angle and investigated whether any GBT-confirmed lines represent potential human disease models. Within the set of GBT-confirmed lines that match a human ortholog (n = 177), 64 (36%) are integrated in genes associated with human diseases, including those of the nervous, circulatory, endocrine, metabolic, digestive, musculoskeletal, immune, and integumentary systems (select genes listed in *Figure 5A*, all 64 genes with disease-associated human orthologs listed in *Supplementary file 3*). 40 of these human disease-associated GBT-confirmed lines represent potential novel genetic disease models as we failed to find a description for any established disease models in mice or zebrafish for orthologs of these genes (*Figure 5B* and *Supplementary file 3*).

## GBT protein trapping creates loss of function products for a diverse population of proteins

Functional genome annotation with GBT constructs is equally powerful in detecting roles for genes in basic cellular processes. We and others have previously used imaging to investigate effective protein trapping in GBT-confirmed lines. This imaging has revealed diverse cellular and subcellular protein expression patterns (*Clark et al., 2011a*; *Ding et al., 2013*; *Ding et al., 2017*; *El-Rass et al., 2017*; *Liao et al., 2012*; *Petzold et al., 2009*; *Westcot et al., 2015*; *Xu et al., 2012*). In this study, we imaged GBT lines using a Lightsheet microscope for the first time. Multi-area tiling with a 20 × objective enabled rapid acquisition of 3-dimensional mRFP fusion protein localization across the entire organism (*Figure 6A*). Confocal imaging in areas of interest revealed even more detail with subcellular resolution (*Figure 6A–B*). Further confocal imaging demonstrated diverse subcellular localizations in GBT-confirmed lines, (*Figure 6B–C*) GBT-candidate lines, and GBT lines (*Figure 6—figure supplement 1*). This subcellular protein localization data from GBT lines can provide crucial information in piecing together gene function.

We therefore wanted to assess the subcellular diversity of all gene products trapped in our current collection of GBT-confirmed lines. As an approach to complement our imaging assessments, we focused on computational approaches to explore subcellular protein diversity in current GBT-confirmed lines. 177 of the GBT-trapped genes were annotated to their human orthologs in at least one public database (ZFIN, Ensembl, Homologene, and InParanoid version 8) (*Supplementary file 1*). Several of these genes were provisionally annotated using BLASTP or a synteny analysis tool,

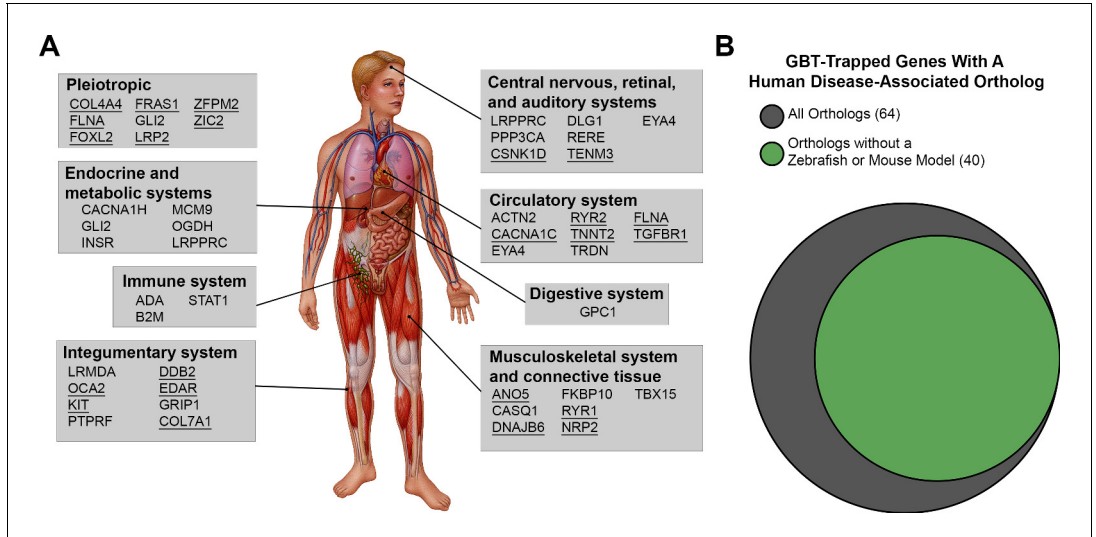

**Figure 5.** Disease-associated human orthologs of the GBT trapped genes are implicated in human genetic disorders of multiple organ systems. (**A**) Representative human orthologs of the GBT-tagged genes are associated with genetic disorders in multi-organ systems. Image provided by Mayo Clinic Media Services. Underline: Disease causative genes with documentations of established disease model in mouse or zebrafish (**B**) Area proportional Venn diagram of 64 human orthologs tagged that are associated with human genetic disorders. 40 human orthologs of GBT-tagged genes are associated with human genetic disorders without an established disease model in zebrafish or mouse. Area proportional Venn diagram was produced using BioVenn (http://www.biovenn.nl/).

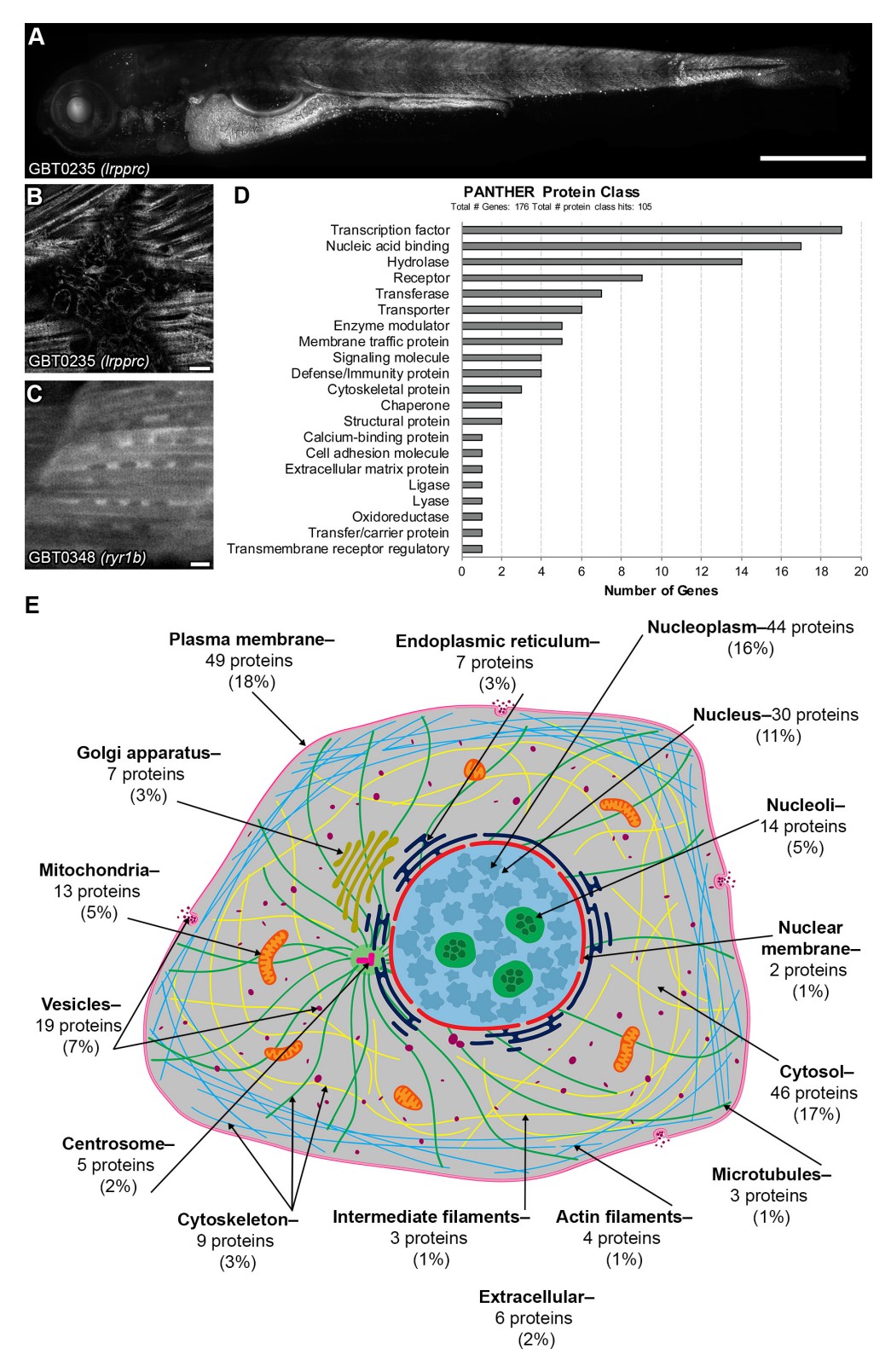

**Figure 6.** GBT-confirmed lines illuminate and disrupt genes encoding proteins with diverse functions and subcellular localizations. The online version of this article includes the following source data and figure supplement(s) for figure 6:

**Source data 1.** PANTHER protein classes of human orthologs tagged in GBT-confirmed lines.

*Figure 6 continued on next page*

Figure 6 continued

**Figure supplement 1.** GBT protein traps illuminate diverse subcellular protein localizations.

SynFind in Comparative Genomics (CoGe) database (https://genomevolution.org/CoGe/SynFind.pl) (*Lyons and Freeling, 2008*). We assessed the functional diversityhuman orthologs of human orthologs of these 177 GBT-confirmed loci with data from the PANTHER version 14.1 database on protein class ontology (http://www.pantherdb.org/) (*Mi et al., 2019*), the Human Protein Atlas on genome-wide experimental proteomics (www.proteinatlas.org. August 27, 2019) (*Uhlén et al., 2015*), and the UniProtKB on knowledge-based proteomics (UniProtKB, https://www.uniprot.org/, *UniProt Consortium, 2018*). PANTHER protein classification revealed that 105 human orthologs (60%, n = 176: see Materials and methods) are classified to at least one of 21 protein classes, and 19 human orthologs (11%) belong to transcription factors (*Figure 6D* and *Figure 6—source data 1*). Human Protein Atlas and UniProtKB subcellular localization data likewise showed diverse classifications of expression with a large group of nuclear localized human orthologs of these 177 GBT-confirmed loci (*Figure 6E* and *Supplementary file 4*).

We then asked whether this if our computational analysis corresponded to the patterns seen in our imaging data. We found that *LRPPRC* (human ortholog of *lrpprc* (GBT0235)—*Figure 6A–B*) was not annotated to a protein class in PANTHER but mapped to mitochondria in Human Protein Atlas, consistent with its puncta expression pattern (*Figure 6A–B*). *RYR1* (human ortholog of *ryr1b* (GBT0348)—*Figure 6C*) was annotated as a transporter in PANTHER and was mapped to the cytosol, Golgi apparatus, and vesicles in Human Protein Atlas, consistent with its more uniform expression pattern (*Figure 6C*). Overall, protein class ontology and known subcellular localizations of cloned GBT genes suggest that the GBT system traps and enables functional annotation for a rich diversity of proteins. Additionally, the GBT-confirmed lines in orthologs of human genes without a known subcellular localization potentiate the discovery of their subcellular expression pattern in the context of a living animal.

## mRFP expression profiling in GBT-confirmed lines reveals substantial new expression data at both 2 dpf and 4 dpf

We next asked if the mRFP expression patterns in our GBT-confirmed lines unveiled novel cellular expression data. To address this, we focused on the GBT-confirmed lines that were non-redundant and mapped to a known protein coding gene. Importantly, these GBT-confirmed lines exhibited expression patterns that are tissue specific and include assorted brain regions, heart, skin, muscle, vasculature, and blood (*Figure 7A–R*). We analyzed publicly available expression data of these 193 tagged genes in wild-type fish (downloaded from ZFIN on August 28th, 2019). Our GBT-confirmed lines revealed expression patterns (available on www.zfishbook.org) for 67 genes at 2 dpf and 174 genes at four dpf without publicly available expression data in ZFIN (*Figure 7S–T*).

## Discussion

### Gene-break transposon system as a next generation mutagenesis system

GBT technology represents the first method for revertible allele generation in vertebrates outside of the mouse model (*Clark et al., 2011a*). In this manuscript we broadened GBT genomic coverage through the development of an RP2 construct series to encode functional mRFP in each of the potential reading frames (*Figure 1A*). While each individual construct still only integrates in-frame in a subset of introns, the RP2 series potentiates in-frame mRFP for any intron with Tol2-mediated integration. We also desired to increase GBT utility for subsequent genomic engineering applications. As RP2 series constructs were not modularly designed, we iteratively developed a next generation RP8 GBT series. All RP8 constructs include new restriction enzyme sites that render them modular for custom engineering. Additionally, RP8 series constructs use a smaller backbone designed to enhance transgenic efficiency. RP8 vectors most notably possess a new 3' exon trap cassette that uses the γ-crystalline promoter to drive expression of lens-specific tagBFP instead of the ubiquitous expression of GFP delivered from RP2 vectors (*Figure 1B*). We found this lens-specific tagBFP to be

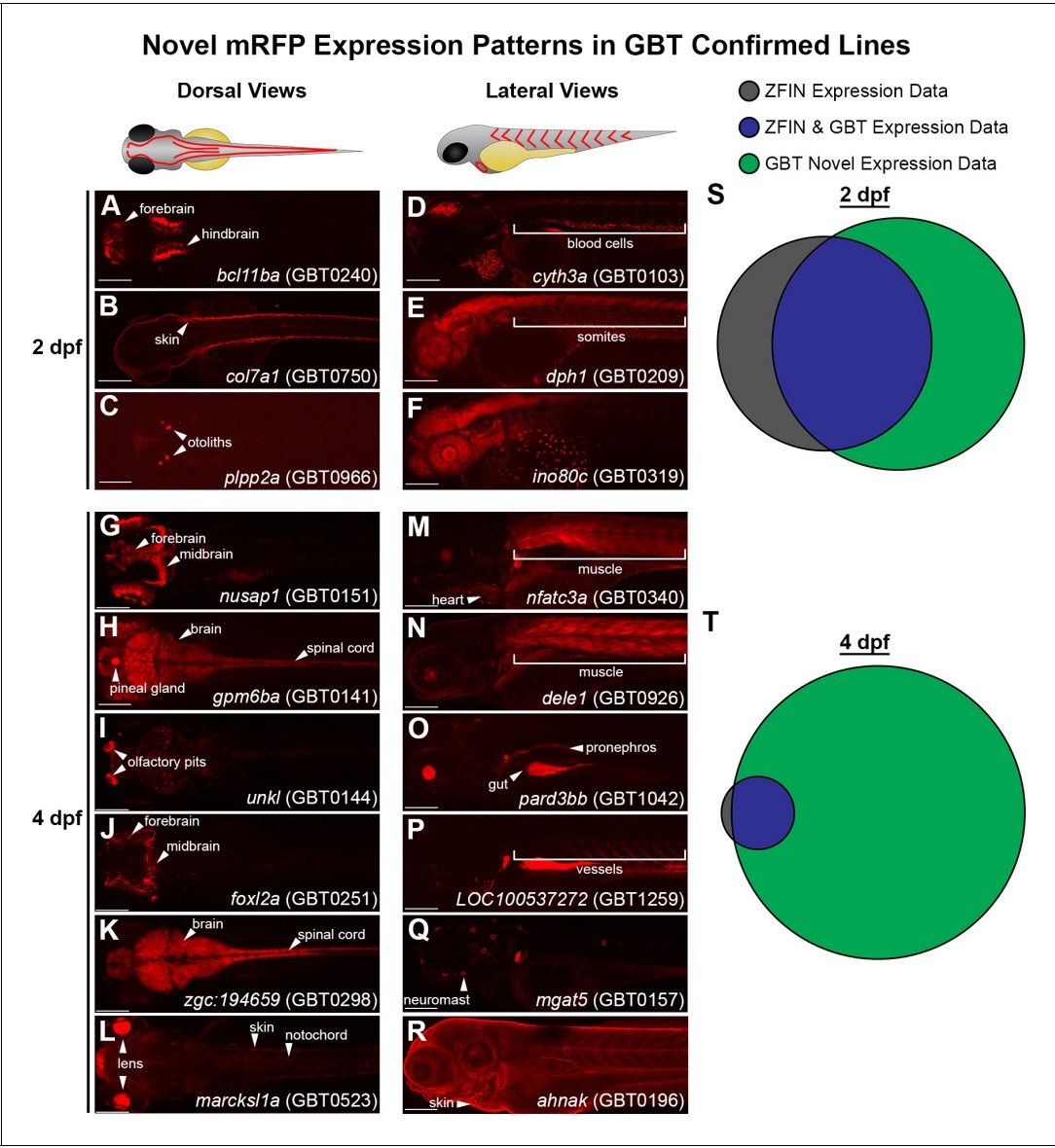

**Figure 7.** GBT protein trap elucidates novel gene expression patterns in embryonic and larval zebrafish. (A–C) Dorsal views of 2 days post-fertilization (dpf) embryos with GBT protein trap mRFP expression patterns ranging from *bcl11ba* in the forebrain and hindbrain (A), to *col7a1* in the skin (B), and *plpp2a* in the otoliths (C). (D-F) Lateral views of 2 dpf embryos with GBT protein trap mRFP expression patterns ranging from *cyth3a* in blood cells (D), to *dph1* in somites (E), and *ino80c* around the yolk (F). (G–L) Dorsal views of GBT protein trap mRFP expression patterns in 4 dpf larvae including *nusap1* in the forebrain and midbrain (G), *gpm6ba* in the brain, spinal cord, and pineal gland (H), *unkl* in the olfactory pits (I), *foxl2a* in the forebrain and midbrain (J), *zgc:194659* in the brain and spinal cord (K), and *marcksl1a* in the lens, skin, and notochord (L). (M–R) Lateral views of GBT protein trap mRFP expression patterns in 4 dpf larvae including *nfatc3a* in heart and muscle (M), *dele1* in muscle (N), *pard3bb* in the gut and pronephros (O), *LOC100537272* in vessels (P), *mgat5* in neuromasts (Q), and *ahnak* in skin (R). Scale bars = 200 µm. (S–T) Area proportional Venn diagrams of 193 genes trapped in GBT-confirmed lines comparing the ZFIN-assembled database with mRFP expression in GBT lines available through zfishbook at two dpf (S) and four dpf (T). 67 (35%) and 174 (90%) of 193 genes trapped in GBT-confirmed lines have no description about wild-type expression at 2 dpf and 4 dpf, respectively.

equally useful in screening founders with GBT integrations. While we did not explicitly validate that the RP8 series provides transcriptional effects equivalent to the RP2 series, the major functional change in RP8 lies in its 3' exon trap. We therefore expect the knockdown to be similar between RP8 and RP2.

Two additional zebrafish transposon-based protein trap vectors have been established. Dr. Fraser's group developed the FlipTrap system (***Trinh et al., 2011***) that is mutagenic in the presence of

Cre-recombinase. However, this FlipTrap system is primarily focused on imaging fusion proteins in vivo and addressing cellular dynamics. The Chen lab developed a complementary flipping system called the FT1 system that uses either Cre or Flp recombinase to regulate its alleles depending on the original insertion orientation (*Ni et al., 2012*). Our GBT system is highly complementary and non-redundant with these alternative transposon-based protein trap methods. The 5' protein trap in the GBT system is terminated through an enhanced polyadenylation signal in conjunction with a putative border element and a second splice acceptor in the 3' exon trap helps eliminate any pass-through. Together these elements achieve higher gene-specific mRNA knockdown than the single splice acceptor and the basic polyadenylation signal in FlipTrap and FT1 (*Figure 3*). In addition to a 5' protein trap, our GBT system also possesses a 3' exon trap that serves as a means for screening integrations, reports 3' sequence, and possesses the ability to trap (without mRFP expression) non-coding RNAs that undergo splicing (*Figure 1*).

Consistent with its mRNA knockdown abilities, the mutagenic efficacy of the GBT system is functionally similar to other genome-wide forward genetic approaches at identifying critical early developmental loci. The GBT system achieved 7% recovery of visible early (through five dpf) developmental phenotypes during an initial forward genetic screen. This recovery is comparable to the 5% recovered visible phenotypes from the Sanger TILLING consortium analysis of truncated zebrafish genes (*Kettleborough et al., 2013*). Our 7% phenotype recovery is also comparable to prior retroviral (*Amsterdam and Hopkins, 2004*) and ENU (*Haffter et al., 1996*) zebrafish mutagenesis works that estimated between 1400 and 2400 genes (~5–9% of the genome) would result in a visible embryonic phenotype when mutated.

## Gene-break transposon system enables functional genome annotation and generates novel potential human disease models

The connections between gene, expression pattern, function, and phenotype (or human disease) can be elucidated using our GBT system. During the initial GBT library creation, we identified a GBT-confirmed line with an RP2 integration between exon 81 and exon 82 of *ryr1b* (ENS-DART00000036015.9, Ensembl Release 100 on April 2020), a zebrafish ortholog of human *RYR1*. Mutations in *RYR1* are well-linked to a rare genetic neuromuscular disorder known as central core disease that presents with mild to severe muscle weakness (*Jungbluth et al., 2018*). Indeed, homozygous animals in this *ryr1b* GBT-confirmed line possess skeletal muscle-specific mRFP expression and a slow swimming phenotype (*Clark et al., 2011a*). Previously, a spontaneous mutant called *relatively relaxed* (*ryr1b^{mi340/mi340}*) was shown to have a slow swimming phenotype, truncated ryr1b protein with a pre-mature stop involved in an insertional mutagen, and defective skeletal muscle Ca$^{2+}$ transients (*Hirata et al., 2007*). Our *ryr1b* GBT-confirmed line was therefore ideal to validate the functional genome annotation abilities of GBT constructs.

Similar to the *relatively relaxed* mutants which carry an insertion that introduces a premature stop codon between exons 48 and 49 of *ryr1b* (*Hirata et al., 2007*), we noted severely dampened skeletal muscle Ca$^{2+}$ transients in animals homozygous for the *ryr1b* GBT allele (*Figure 4*). In addition to previously reported decreases in peak amplitude (*Hirata et al., 2007*) we found that *ryr1b^{mn0348Gt/mn0348Gt}* animals also displayed a slower upstroke and lower frequency of skeletal muscle Ca$^{2+}$ transients than wildtypes, functionally annotating these roles for the C-terminal region of *ryr1b* gene in vivo. Including this *ryr1b* line, we generated GBT-confirmed lines with integrations in 64 zebrafish orthologs of human disease-associated genes (*Figure 5*, *Supplementary file 1*, *Supplementary file 3*) in this study. GBT-confirmed lines with integrations in 40 zebrafish orthologs of human disease-associated genes may represent novel potential disease models, as we failed to find any description of existing zebrafish or mouse models for these genes or diseases. Our GBT system importantly possesses a built-in cure due to its revertible nature. Therefore, these GBT potential disease models will allow direct comparison of tissue-specific gene restoration with any therapeutic approach.

## GBT protein trapping provides the basis for annotation of functionally diverse proteins and novel transcripts mapped on poorly assembled genomic regions

To achieve genomic representation, unbiased protein trapping is an important consideration. Tol2 transposase-mediated systems are known to facilitate near-random integration, but we wanted to

explore this in the context of the GBT protein trapping constructs. We utilized PANTHER, Human Protein Atlas, and UniProtKB to explore the protein class and subcellular localization of the human orthologs of the genes trapped in GBT-confirmed lines (*Figure 6*). While nuclear localized proteins, such as transcription factors, represented the largest class of GBT-trapped genes, we identified a diverse set of proteins in our GBT-confirmed lines that localize all the way from the nucleoli to the extracellular space. The reason for the enrichment of nuclear genes is unknown. The rich diversity of proteins observed in our GBT-confirmed lines still supports that the entire collection has high diversity and is consistent with the random nature of Tol2-mediated genome integration events (*Clark et al., 2011a*). Completion of the zebrafish reference genomes also has enabled many new discoveries to be made with regards to the position of hundreds of genes that affect embryogenesis, behavior, and physiology. However, poorly assembled regions remain in both the zebrafish and the human genome (*Howe et al., 2013*). We indeed found that 10 GBT integrations in the confirmed lines (with mRFP expression) failed to map to any predicted genes. However, RNA sequencing reads in public datasets identified potential unannotated coding sequences aligned with these GBT integration loci (*Supplementary file 1*). While 5' and 3' RACE are necessary to confirm the mRNA fusion products, these unannotated coding sequences represent the possibility to annotate novel, protein-coding transcripts in these GBT lines. Therefore, GBT protein trapping can find, illuminate expression, and elucidate in vivo functions of novel genes and/or gene variants in poorly annotated regions of reference genomes.

## GBT protein trapping annotates novel endogenous gene expression

GBT-based mRFP fusion proteins represent a notable advance over traditional techniques for probing endogenous gene expression (e.g., immunohistochemistry, in situ hybridization) that have yielded very little gene expression data at later developmental stages. The truncated mRFP fusion proteins in both RP2 and RP8 series constructs exhibited distinct cellular localizations in our GBT lines throughout development, including two dpf and beyond. Approximately 90% of GBT-confirmed lines showcased novel expression patterns of their annotated genes at four dpf (*Figure 7*).

These GBT-based mRFP fusion proteins allow investigation of subcellular localization of these tagged-gene products (*Figure 6*, *Figure 6—figure supplement 1*), with the exception of cases where the protein localization signal is contained in the C-terminal domain (*Clark et al., 2011a*; *Trinh and Fraser, 2013*). We indeed observed mRFP accumulation in the kidney tubules, white blood cells or developing bones in some GBT lines, likely based upon the remaining signal sequences at the N-terminus of the endogenous protein. Still, visualizing protein expression dynamics of GBT-trapped proteins in most lines should facilitate important initial annotations regarding subcellular localization of uncharacterized proteins to investigate molecular functions in vivo. With ever-improving fluorescence-based imaging tools (*Liu et al., 2018*), our GBT lines have the potential to annotate both cellular and subcellular gene expression at diverse stages on an organismal scale.

## Gene-break protein trap library is a rich resource for the community

Taken together, GBT-based mRFP-reporters demonstrate how much we still have left to understand about the expression patterns of the overall proteome and, ultimately, the complex codex that is our genome. Even at the relatively well-studied 2-dpf stage, nearly 40% of GBT-confirmed lines elucidated novel gene expression data (*Figure 7*). Cataloging these expression patterns enables investigators to make collections of lines with expression in their cell/tissue of interest and/or a phenotype. The remaining 144 GBT-candidate lines and over 800 GBT lines represent a rich resource for genomic discoveries. For any GBT-candidate or GBT lines of interest, a similar cloning pipeline (*Figure 2*) can be employed to identify the GBT integration locus. In addition, the refinement of the zebrafish genome will enhance our ability to complete the annotation from GBT-line to GBT-confirmed line for any given line with a desired expression profile and/or phenotype. Together, this 1,200+ GBT-line collection is a new contribution for using zebrafish to annotate the vertebrate genome.

## Future genomic insights using the GBT system

Although our GBT lines were made with random integration, new targeted integration tools, such as GeneWeld (*Wierson et al., 2020*), that employ gene editing techniques will empower labs to build custom GBT lines for their gene of interest (*El Khoury et al., 2018*). The three reading frames and

modularity of the RP8 series are especially well suited to targeted integration approaches. Further, a combination of targeted and random integration may best facilitate discovery. For instance, a targeted approach could integrate a GBT cassette into a well-characterized, process-associated gene. Then, random integration could be used to probe for genes that potentiate or abrogate the disruption in the original process-associated gene. With this approach, GBTs are a powerful tool to investigate multigenic processes, including human disease.

## GBT system offers advantages over frame-shift mutations

Finally, targeted mutagenic technology, such as CRISPR and TALEN systems, has become the gold standard reverse genetic approach. However, engineered mutant animals using approaches generate targeted indel mutation frequently fail to display overt phenotypes, often explained by genetic compensation (*Balciunas, 2018*; *El-Brolosy and Stainier, 2017*). One mechanism includes cellular increases in transcripts of genes in the same family that can functionally substitute when activated in a mutant background (*Balciunas, 2018*). Recently, a number of studies using reverse genetics tools have revealed phenotype differences between knockouts (indel mutants), and knockdowns (antisense-treated animals) in multiple model systems including *Arabidopsis*, *Drosophila*, zebrafish, mouse, and human cell lines. This discrepancy is attributed to transcriptomic changes in mutant but not in knockdown animals (Reviewed from *El-Brolosy and Stainier, 2017*). For example, knockdown of *egfl7*, an endothelial extracellular matrix (ECM) gene, induces severe vascular defects, whereas most *egfl7* mutants exhibit no obvious defect, resulting from upregulation of other ECM proteins, especially emilins in *egfl7* mutants, but not in *egfl7* morphants (*Ronzitti, 2019*). As another mechanism of genetic compensation, mRNA processing—including nonsense-associated exon skipping and the use of alternative start or splice sites to escape nonsense-mediated decay—has been recently demonstrated to hinder loss-of-function approaches in zebrafish (*Anderson et al., 2017*; *Prykhozhij et al., 2017*), in human cell lines (*Lalonde et al., 2017*; *Winter et al., 2019*) and in the human population (*Jagannathan and Bradley, 2016*). In contrast, the molecular mechanism of GBT mutagenesis can normally avoid this genetic compensation effect seen with small indel mutations because the strong poly (A)-trapping element in the 5' exon trap domain of RP cassettes can reduce mRNA to below 1% of the complete wild-type transcript level. This reduction eliminates many sources of transcriptional adaptations triggered by a loss of function mutation, such as alternative transcriptional start sites, splicing, or alternative translation initiation. The GBT can therefore act as a useful validation tool when targeted mutations with other technologies fail to display any phenotype.

# Materials and methods

**Key resources table**

| Reagent type (species) or resource | Designation | Source or reference | Identifiers | Additional information |
|---|---|---|---|---|
| Recombinant DNA reagent | pGBT-RP2.1 | *Clark et al., 2011a* | RRID:Addgene_31828, Genbank: HQ335170 | *Figure 1A* |
| Recombinant DNA reagent | pGBT-RP2.2 | This paper | Genbank: MT815588 | *Figure 1A* |
| Recombinant DNA reagent | pGBT-RP2.3 | This paper | Genbank: MT815589 | *Figure 1A* |
| Recombinant DNA reagent | pGBT-RP8.1 | This paper | Genbank: MT815590 | *Figure 1B* |
| Recombinant DNA reagent | pGBT-RP8.2 | This paper | Genbank: MT815591 | *Figure 1B* |
| Recombinant DNA reagent | pGBT-RP8.3 | This paper | Genbank: MT815592 | *Figure 1B* |
| Recombinant DNA reagent | pGBT-RP7.1 | This paper | | An intermediate construct to create pGBT-RP8.1 |

*Continued on next page*

*Continued*

| Reagent type (species) or resource | Designation | Source or reference | Identifiers | Additional information |
|---|---|---|---|---|
| Recombinant DNA reagent | pGBT-RP6.1 | This paper | | An intermediate construct to create pGBT-RP8.1 |
| Recombinant DNA reagent | pGBT-RP5.1 | This paper | | An intermediate construct of pGBT-RP8.1 |
| Recombinant DNA reagent | pre(−1)GBT-RP5.1 | This paper | | An intermediate construct of pGBT-RP5.1 |
| Recombinant DNA reagent | pre(−2)GBT-RP5.1 | This paper | | An intermediate construct of pGBT-RP5.1 |
| Recombinant DNA reagent | pre(−3)GBT-RP5.1 | This paper | | An intermediate construct of pGBT-RP5.1 |
| Recombinant DNA reagent | pKTol2-SE | *Clark et al., 2011b* | | |
| Recombinant DNA reagent | pUC57-I-SceI_LoxP_Splice | This paper | | DNA source to create pre(−3)GBT-RP5.1 |
| Recombinant DNA reagent | pUC57 | Genscript | SD1176 | |
| Recombinant DNA reagent | pKTol2gC-nlsTagBFP | This paper | | DNA source to create pre(−2)GBT-RP5.1 |
| Recombinant DNA reagent | pGBT-R15 | *Clark et al., 2011a* | RRID:Addgene_31826, Genbank ID: HQ335168 | |
| Recombinant DNA reagent | pGBT-PX | *Sivasubbu et al., 2006* | RRID:Addgene_31824, Genbank ID: HQ335166 | |
| Recombinant DNA reagent | pCR4-bactmIntron | This paper | | DNA source to create pGBT-RP8.1 |
| Recombinant DNA reagent | pCR4-bact_I1 | This paper | | DNA source of the carp beta-actin intron amplified from pGBT-RP2.1 |
| Recombinant DNA reagent | pCR4-TOPO | Invitrogen | 450030 | |
| Recombinant DNA reagent | pEXPR-mylpfa:GCaMP3 | *Baxendale et al., 2012* | | |
| Chemical compound, drug | phenylthiocarbamide | Sigma-Aldrich | P7629 | |
| Chemical compound, drug | tricaine | Sigma-Aldrich | A5040 | |
| Chemical compound, drug | low melt agarose | Fisher Scientific | BP1360 | |
| Chemical compound, drug | pentylenetetrazole | Sigma-Aldrich | P6500 | |
| Chemical compound, drug | (S)-(-)-blebbistatin | Tocris | 1852 | |
| Chemical compound, drug | β-mercaptoethanol | Sigma-Aldrich | M6250 | |
| Chemical compound, drug | proteinase K | Roche | 3115879001 | |
| Commercial assay or kit | T4 DNA ligase | New England Biolabs | M0202S | |
| Commercial assay or kit | RNeasy Micro Kit | QIAGEN | 74004 | |
| Commercial assay or kit | Stainless steel beads | Next Advance | SSB05 | |
| Commercial assay or kit | MaXtract High Density tubes | QIAGEN | 129056 | |

*Continued on next page*

*Continued*

| Reagent type (species) or resource | Designation | Source or reference | Identifiers | Additional information |
|---|---|---|---|---|
| Commercial assay or kit | SuperScript II Reverse Transcriptase | Thermo Fisher Scientific | 18064014 | |
| Commercial assay or kit | SensiFAST SYBR Lo-ROX kit | Bioline | BIO-94005 | |
| Commercial assay or kit | QIAquick Gel Extraction Kit | QIAGEN | 28704 | |
| Software, algorithm | GraphPad Prism 8 | GrapgPad | RRID:SCR_002798 | |
| Software, algorithm | R | www.R-project.org | RRID:SCR_001905 | |
| Software, algorithm | R-Studio | www.rstudio.com/ | | |
| Software, algorithm | pwr package | https://CRAN.R-project.org/package=pwr | | |
| Software, algorithm | wilcox.test function | www.R-project.org | | |
| Software, algorithm | coin package | *Hothorn et al., 2006*, *Hothorn et al., 2008* | | |
| Software, algorithm | outliers package | https://CRAN.R-project.org/package=outliers | | |
| Software, algorithm | effsize package | https://CRAN.R-project.org/package=effsize | | |
| Software, algorithm | JMP version 14 | http://www.jmp.com/en_us/software/jmp.html | RRID:SCR_014242 | |
| Software, algorithm | SynFind | https://genomevolution.org/CoGe/SynFind.pl | | |
| Software, algorithm | BLASTP | http://blast.ncbi.nlm.nih.gov/Blast.cgi?PROGRAM=blastp&PAGE_TYPE=BlastSearch&LINK_LOC=blasthome | RRID:SCR_001010 | |
| Software, algorithm | BioMart, Ensembl tool | http://useast.ensembl.org/biomart/martview/ | RRID:SCR_002344 | |
| Software, algorithm | PANTHER version 14.1 | http://www.pantherdb.org/ | RRID:SCR_004869 | |
| Software, algorithm | FIJI | https://fiji.sc/ | RRID:SCR_002285 | |
| Software, algorithm | MetaMorph Microscopy Automation and Image Analysis Software | Molecular Devices | RRID:SCR_002368 | |
| Software, algorithm | Digidata 1440A | Molecular Devices | | |
| Software, algorithm | Clampex 10.3 | Molecular Devices | | |
| Software, algorithm | Integrative Genomics Viewer (version 2.4.19) | *Thorvaldsdóttir et al., 2013* | RRID:SCR_011793 | |
| Software, algorithm | Galaxy | https://usegalaxy.org/ | RRID:SCR_006281 | |
| Software, algorithm | BAMtools | *Barnett et al., 2011* | RRID:SCR_015987 | |
| Software, algorithm | TopHat | *Kim et al., 2013* | RRID:SCR_013035 | |
| Software, algorithm | Zebrafish Information Network (ZFIN) | https://zfin.org/ | RRID:SCR_002560 | |
| Software, algorithm | Ensembl | https://useast.ensembl.org/index.html | RRID:SCR_002344 | |
| Software, algorithm | InParanoid version 8 | http://inparanoid.sbc.su.se/cgi-bin/index.cgi | RRID:SCR_006801 | |

*Continued on next page*

*Continued*

| Reagent type (species) or resource | Designation | Source or reference | Identifiers | Additional information |
|---|---|---|---|---|
| Software, algorithm | The Human Protein Atlas | www.proteinatlas.org | RRID:SCR_006710 | |
| Software, algorithm | UniProtKB | https://www.uniprot.org/ | RRID:SCR_004426 | |
| Software, algorithm | Online Mendelian Inheritance in Man (OMIM) | https://omim.org/ | RRID:SCR_006437 | |
| Software, algorithm | Mouse Genome Informatics (MGI) | http://www. informatics.jax.org | RRID:SCR_006460 | |
| Software, algorithm | zfishbook | https://zfishbook.org/ | RRID:SCR_006896 | |
| Other | RNA-seq dataset | *White et al., 2017* | GRCz10.WTSI. 36hpf.1.bam | ftp://ftp.ensembl.org/pub/ data_files/danio_rerio/ GRCz10/rnaseq/ |
| Other | RNA-seq dataset | *White et al., 2017* | GRCz10.WTSI. 48hpf.1.bam | ftp://ftp.ensembl.org/pub/ data_files/danio_rerio/ GRCz10/rnaseq/ |
| Other | RNA-seq dataset | *White et al., 2017* | GRCz10.WTSI. 4dpf.1.bam | ftp://ftp.ensembl.org/pub/ data_files/danio_rerio/ GRCz10/rnaseq/ |

## Zebrafish husbandry

All zebrafish (*Danio rerio*) were maintained according to the procedures described previously (*Leveque et al., 2016*).

## Generating GBT constructs, RP2 and RP8 series

pGBT-RP8.2 and -RP8.3 were made by combining three restriction endonuclease fragments of pGBT-RP8.1, a 2.2 kb AflII to AgeI, a 0.7 kb EcoRI to SpeI, and a 3.0 kb SpeI to AflII, with a short adapter to close the space between AgeI and EcoRI that effectively removed one or two thymine nucleotides just following the splice acceptor prior to the AUG-less mRFP cassette. For pGBT-RP8.2, Adapter-GBT(+2) was made by annealing oligos adapter-GBT(+2)-a [CCGGTTTTCTCATTCATTTA-CAGTCAGCCGG] and adapter-GBT (+2)-b [AATTCCGGCTGACTGTAAATGAATGAGAAAA]. For pGBT-RP8.3, Adapter-GBT(+3) was made by annealing oligos adapter-GBT (+3)-a [CCGGTTTTCTCA TTCATTTACAGCAGCCGG] and adapter-GBT(+3)-b [AATTCCGGCTGCTGTAAATGAATGAGAAAA ].

pGBT-RP2.2 and -RP2.3 were made by combining three restriction endonuclease fragments of pGBT-RP2.1 (*Clark et al., 2011a*), a 3.6 kb BlpI to AgeI, a 1.9 kb EcoRI to AvrII, and a 3.55 kb AvrII to BlpI, with a short adapter to close the space between AgeI and EcoRI that effectively removed one or two thymine nucleotides just following the splice acceptor prior to the AUG-less mRFP cassette. For pGBT-RP2.2, Adapter-GBT(+2) was made by annealing oligos adapter-GBT(+2)-a and adapter-GBT(+2)-b. For pGBT-RP2.3, Adapter-GBT(+3) was made by annealing oligos adapter-GBT (+3)-a and adapter-GBT(+3)-b. pGBT-RP8.1 was made by cloning a mini-intron derived from carp beta actin intron one into pGBT-RP7.1. The 234 bp SalI to XhoI mini-intron fragment was isolated from pCR4-bactmIntron following digestion. The pGBT-RP7.1 plasmid was digested with XhoI so that the SalI to XhoI fragment was cloned between the gamma-crystallin promoter and nls tagBFP.

pCR4-bactmIntron was made by removing a 1.1 kb internal portion of the carp beta actin intron one by digestion of pCR4-bact_I1 with BstBI and BssHII, followed by filling in 5' overhangs and ligating remaining vector fragment.

pCR4-bact_I1 was cloning a PCR product containing the carp beta-actin intron into pCR4-TOPO (450030, Invitrogen, Thermo Fisher Scientific, Waltham, MA). The intron was amplified from pGBT-RP2.1 (*Clark et al., 2011a*) using MISC-bact_exon-F1 [CAGCTAGTGCGGAATATCATCTGCC] and MISC-bact_intron-R1 [CTTCTCGAGGTGAATTCCGGCTGAACTGTA] primers.

pGBT-RP7.1 was made by replacing a 501 bp PstI to PstI fragment of pGBT-RP6.1 with a 480 bp PstI to PstI fragment of pRP2.1. This changed the nucleotide sequence between the carp beta-actin

splice acceptor to replicate the sequences in pGBT-RP2.1. pGBT-RP7.1 was never directly tested in zebrafish.

pGBT-RP6.1 was made by flipping the internal trap cassette relative the Tol2 inverted terminal repeats in pGBT-RP5.1. To do this, pGBT-RP5.1 was cut with EcoRV and SmaI. The 2.27 kb EcoRV to SmaI vector backbone fragment, which included the ITRs, was ligated to the 3.51 kb EcoRV to SmaI trap fragment. pGBT-RP6.1 was then selected based on the right ITR of Tol2 being in front of the RFP trap, which is the same orientation of pGBT-RP2.1.

pGBT-RP5.1 was made by cloning a PCR product with the AUG-less mRFP into pre(−1)GBT-RP5.1. The 698 bp mRFP* PCR product was obtained by amplification of pGBT-R15 (*Clark et al., 2011a*) with CDS-mRFP*-F1 [AAGAATTCGAAGGTGCCTCCTCCGAGGATGTCATCAAGG] and CDS-mRFP-R1 [AAACTAGTCTTAGGCTCCGGTGGAGTGGCGG]. Prior to cloning the PCR mRFP* product was digested with EcoRI and SpeI to prepare the ends for subcloning into pre(−1)GBT-RP5.1 that was opened between the carp beta actin splice acceptor and the ocean pout terminator.

pre(−1)GBT-RP5.1 was made by cloning 1.2 kb SpeI to AvrII fragment from pGBT-PX (*Sivasubbu et al., 2006*) that contained the ocean pout terminator into the SpeI site of pre(−2)GBT-RP5.1. The resulting products were screened for the proper orientation of the ocean pout terminator relative to the carp beta actin splice acceptor.

pre(−2)GBT-RP5.1 was made by inserting an expression cassette to make a 3' poly(A) trap that makes blue lenses. A 1.15 kb SpeI to BglII fragment from pKTol2gC-nlsTagBFP was cloned into pre(−3)GBT-RP5.1 that had been cut with AvrII and BglII. This moved the *Xenopus* gamma crystallin promoter driving a nuclear-localized TagBFP in front of the carp beta actin splice donor within pre(−3)GBT-RP5.1 to create a localized BFP poly(A) trap signal replacing the ubiquitous GFP signal that was in pGBT-RP2.1.

pre(−3)GBT-RP5.1 was made by cloning a 492 bp XmaI to NheI scaffold fragment from pUC57-I-SceI_loxP_splice into pKTol2-SE (*Clark et al., 2011b*) opened with XmaI and NheI.

pUC57-I-SceI_LoxP_Splice contains a synthetic sequence (see below) cloned into pUC57 (SD1176, Genscript, Piscataway, NJ). The scaffold contains an I-SceI site; loxP site; carp beta actin splice acceptor; cloning sites for mRFP, ocean pout terminator, and BFP lens cassettes; carp beta actin splice donor; loxP site; and an I-SceI site.

The synthetic sequence described above is:
cccgggatagggataacagggtaatataacttcgtatagcatacattatacgaagttat
cgttaccacccactagcggtcagactgcagattgcagcacgaaacaggaagctgac
tccacatggtcacatgctcactgaagtgttgacttccctgacagctgtgcactttctaaa
ccggttttctcattcatttacagttcagcctgttacctgcactcaccgacaagctgttacc
ctggaattcgtttaaacactagtcaccggcgttcctaggttataagatctacctaaggtg
agttgatctttaagcttttacattttcagctcgcatatatcaattcgaacgtttaattagaat
gtttaaataaagctagattaaatgattaggctcagttaccggtctttttttttctcatttacact
gagctcaagacgtctgataacttcgtatagcatacattatacgaagttattaccctgttatccctatggctagc.

## Generating GBT collection

Generation of the GBT collection was based on the prior described protocols (*Clark et al., 2011a*; *Ni et al., 2016*). RP2 constructs were injected and sorted by low mosaicism of GFP expression. RP8 constructs were sorted on the basis of strong tagBFP expression in the eyes. Overall ~30% of injected fish met these criteria. ~ 25% of these F0 fish gave RFP offspring.

## Fluorescent microscopy of mRFP reporter protein expression

Larvae were treated with 0.2 mM phenylthiocarbamide (P7629, Sigma-Aldrich, St. Louis, MO) at one dpf to inhibit pigment formation. The anesthetized fish were mounted in 1.5% low-melt agarose (BP1360, Fisher Scientific, Hampton, NH) prepared with 0.017 mg/ml tricaine (Ethyl 3-aminobenzoate methanesulfonate salt, A5040, Sigma-Aldrich) solution in an agarose column in the imaging chamber. The protocol of ApoTome microscopy was described in previous publication. (*Clark et al., 2011a*) For Lightsheet microscopy, larval zebrafish were anesthetized with 0.017 g/ml tricaine in 1.5% low-melt agarose (BP1360, Fisher Scientific) and mounted in glass capillaries. To capture RFP expression patterns of 2 dpf and four dpf larval zebrafish, LP 560 nm filter as excitation and LP 585 nm as emission was used for Lightsheet microscopy. The sagittal-, dorsal-, and ventral- oriented

z-stacks of the mRFP expression were captured at either 50x magnification using an ApoTome microscope (Zeiss, Oberkochen, Germany) with a 5x/0.25 NA dry objective (Zeiss) or 50x magnification using a Lightsheet Z.1 microscope (Zeiss) 5x/0.16 NA dry objective. Each set of images were obtained from the same larva and the images shown are composites of the maximum image projections of the z-stacks obtained from each direction.

For confocal microscopy, larval zebrafish were anesthetized with 0.017 g/ml tricaine (Ethyl 3-aminobenzoate methanesulfonate salt, A5040, Sigma-Aldrich) in 1.0% low-melt agarose (BP1360, Fisher Scientific) and mounted 35 mm glass-bottom dishes (P35G-1.5–14 C, MatTek Life Sciences, Ashland, MA). Imaging was performed on an LSM-780 (Zeiss, Oberkochen, Germany) using either a C-Apochromat 63x/1.2 NA or a C-Apochromat 40x/1.2 NA water immersion objective. RFP was excited at 561 nm and emissions 570–750 nm were collected.

## Sperm cryopreservation

Sperm collection and cryopreservation was initially based on the protocol described in *Draper and Moens, 2009* and moved to the Zebrafish International Resource Center (ZIRC) protocol described in *Matthews et al., 2018*.

## Genomic DNA isolation

Genomic DNA was isolated from F2 fish tail biopsies to conduct next generation sequencing and from both wild-type and heterozygous larva to manually perform PCR-based analysis for the identification of GBT-insertion site. 60 µl of lysis buffer containing with 10 mM Tris (pH 8.0), 100 mM NaCl, 10 mM EDTA, 0.4% SDS and 5 µg/ml proteinase K (03115879001, Roche) was loaded into each well of a 96-wells plate and clipped adult fins/larvae were individually placed into each well and incubated at 50℃ for 3 hr. The solution with lysed tissue were suspended with a multichannel micropipette to dissolve the tissue, mixed with 60 µl of isopropanol and centrifuged at ~3,000 rpm/ 15,000 × g for 20 min at 4℃. After removing the supernatant, 100 µl of 70% ethanol were added, centrifuged for 20 min at 4℃. After discarding ethanol, the pellets were dried and re-suspended in 50 µl of water/TE. As the alternative protocol to quickly extract genomic DNA from zebrafish larvae, the specimens were individually placed to 0.2 ml PCR tubes and lysed with 30–50 µl of 50 mM NaOH and incubated at 95 C° for 20 min. The solutions with lysed specimens were vortexed and neutralized with 1 of 10 vol 1M Tris-HCl.

## Identification of GBT insertion loci

In addition to the broad next gen sequencing approach used to identify GBT integration sites, we used a combination of several a la carte methods including 5' and 3' rapid amplification of cDNA ends (RACE) (*Clark et al., 2011a*), inverse PCR (*Clark et al., 2011a*), and Thermal Asymmetric Interlaced PCR (TAIL-PCR). The protocol used for TAIL-PCR was designed to amplify and clone junction fragments from Tol2-based gene-break transposons was based on a protocol from *Parinov et al., 2004* with some modifications. The following primer mixtures (containing 0.4 µM GBT specific primer and 2 µM degenerate primers (DP)) were prepared: for primary PCR: 5R-mRFP-P1/DP1, 5R-mRFP-P1/DP2, 5R-mRFP-P1/DP3, 5R-mRFP-P1/DP4, 3 R-GM2-P1/DP1, 3 R-GM2-P1/DP2, 3 R-GM2-P1/DP3, 3 R-GM2-P1/DP4, 3R-tagBFP-P1/DP1, 3R-tagBFP-P1/DP2, 3R-tagBFP-P1/DP3, 3R-tagBFP-P1/DP4; for secondary PCR: 5R-mRFP-P2/DP1, 5R-mRFP-P2/DP2, 5R-mRFP-P2/DP3, 5R-mRFP-P2/DP4, 3 R-GM2-P2/DP1, 3 R-GM2-P2/DP2, 3 R-GM2-P2/DP3, 3 R-GM2-P2/DP4, 3R-tagBFP-P2/DP1, 3R-tagBFP-P2/DP2, 3R-tagBFP-P2/DP3, 3R-tagBFP-P2/DP4; for tertiary PCR: TAIL-bA-SA/DP1, TAIL-bA-SA/DP2, TAIL-bA-SA/DP3, TAIL-bA-SA/DP4, Tol2-ITR(L)-O1/DP1, Tol2-ITR(L)-O1/DP2, Tol2-ITR(L)-O1/DP3, Tol2-ITR(L)-O1/DP4, Tol2-ITR(L)-O3/DP1, Tol2-ITR(L)-O3/DP2, Tol2-ITR(L)-O3/DP3, Tol2-ITR(L)-O3/DP4. A total of 1 µl of primer mixtures were added to PCR reaction (total volume 25 µl). Cycle settings were as follows. Primary: (1) 95℃, 3 min; (2) 95℃, 20 s; (3) 61℃, 30 s; (4) 70℃, 3 min; (5) go to 'cycle 2' five times; (6) 95℃, 20 s; (7) 25℃, 3 min; (8) ramping 0.3°/sec to 70℃; (9) 70℃, 3 min; (10) 95℃, 20 s; (11) 61℃, 30 s; (12) 70℃, 3 min; (13) 95℃, 20 s; (14) 61℃, 30 s; (15) 70℃, 3 min; (16) 95℃, 20 s; (17) 44℃, 1 min; (18) 70℃, 3 min; (19) go to 'cycle 10' 15 times; (20) 70℃, 5 min; Soak at 12℃. A total of 5 µl of the primary reaction was diluted with 95 µl of 10 mM Tris-Cl or TE buffers and 1 µl of the mixture was added to the secondary reaction. Secondary: (1) 95℃, 2 min (2) 95℃, 20 s; (3)61℃, 30 s; (4) 70℃, 3 min; (5) 95℃, 20 s; (6) 61℃, 30 s; (7) 70℃, 3 min; (8) 95℃, 20 s;

(9) 44℃, 1 min; (10) ramping 1.5°/sec to 70℃; (11) 70℃, 3 min; (12) go to 'cycle 2' 15 times; (13) 70°C, 5 min; Soak at 12℃. A total of 5 μl of the primary reaction was diluted with 95 μl of 10 mM Tris-Cl or TE buffers and 1 μl of the mixture was added to the tertiary reaction. Tertiary: (1) 95℃, 2 min; (2) 95℃, 20 s; (3) 44℃, 1 min; (3) ramping 1.5°/sec to 70℃; (4) 70℃, 3 min; (5) go to 'cycle 2' 32 times; (6) 70℃, 5 min; Soak at 12℃. Products of the secondary and tertiary reactions were separated by using 1–1.5% agarose gel. The individual bands from the 'band shift' pairs were cut from the gel and purified by using QIAquick Gel Extraction Kit (28704, QIAGEN, Hilden, Germany), and sequenced by the sequencing service in the Medical Genome Facility at Mayo Clinic.

Alternatively, 5' RACE, 3' RACE, or inverse PCR were used to identify the interrupted gene as previously described (*Clark et al., 2011a*).

## Bioinformatic analysis of public RNA sequencing data at the integrated loci

Public datasets of zebrafish wildtype at 36 hpf, 48 hpf and four dpf were downloaded from Ensembl database (Downloaded datasets: GRCz10.WTSI.36hpf.1.bam, GRCz10.WTSI.48hpf.1.bam and GRCz10.WTSI.4dpf.1.bam, URL: ftp://ftp.ensembl.org/pub/data_files/danio_rerio/GRCz10/rnaseq/) (*White et al., 2017*) to browse mapping RNA sequence (RNA-seq) reads around the integration loci. With Galaxy (https://usegalaxy.org/) as a web-based platform for next generation sequencing data analysis (*Afgan et al., 2018*), these downloaded BAM files of these datasets were converted FASTQ file format using BAMtools (*Barnett et al., 2011*). TopHat created a new BAM file and re-aligned RNA-seq reads in the FASTQ file to identify splice junctions between exons in each dataset (*Kim et al., 2013*). These re-mapped BAM files were used to predict candidate transcripts integrated with RP2/RP8 constructs with Integrative Genomics Viewer (version 2.4.19) (*Thorvaldsdóttir et al., 2013*).

## Calcium imaging

Our calcium imaging protocols were modified from *Baxendale et al., 2012*. Calcium data are compiled from two similar methods with independent experimenters and equipment. L.E.G. performed two independent runs of 'Method 1', and A.J.T. performed two independent runs of 'Method 2'. Zebrafish embryos at the single cell (single-cell to 4 cell in Method 1) stage were injected with 80–100 pg (Method 1) or 2 nL of *pEXPR-mylpfa:GCaMP3* plasmid (a gift from Dr. Cunliffe) diluted in water (Method 1) or to 50 ng/μL in 200 mM KCl/0.05% phenol red (Method 2). On day 2, GCaMP3$^+$ embryos were de-chorionated and allowed to rest for 30 min.

For Method 1, embryos were singly incubated in E2 medium containing 20 mM pentylenetetrazole (PTZ) (P6500, Sigma-Aldrich) for approximately 5 min and mounted with a 3% methylcellulose solution containing 20 mM PTZ in a PPT tube and viewed laterally (similar to SCORE imaging [*Petzold et al., 2010*]). After mounting, Ca$^{2+}$ transients in muscles were assessed using an Axio Scope.A1 (Zeiss) equipped with a D3 DSLR camera (Nikon, Minato, Tokyo, Japan) and a HXP 120 fluorescent lamp (Zeiss) using a 10x, 0.45 NA objective. Images were acquired at a rate of 2 Hz over 30 s.

For Method 2, four embryos at a time were transferred into E2 medium containing 5 μM (*S*)-(-)-blebbistatin (1852, Tocris, Bristol, United Kingdom) and incubated for at least 30 min until paralyzed. Once paralyzed, we embedded pairs of embryos into glass bottom dishes (MatTeK Corporation, Ashland, MA) with 1% low melting agarose (BP1360, Fisher Scientific) in E2 embryo medium with 5 μM (*S*)-(-)-blebbistatin and 20 mM PTZ. After an incubation period of 10–20 min, Ca$^{2+}$ transients in muscles were assessed using an inverted IX70 microscope (Olympus, Shinjuku, Tokyo, Japan) equipped with an OrcaFlash4.0 V2 sCMOS camera (Hamamatsu, Hamamatsu City, Shizuoka, Japan) and a pE-300ultra LED light source (CoolLED, Andover, United Kingdom) using a 20x, 0.75 NA objective. Images were acquired for 3 min at 5 Hz as a stream using Metamorph software (Molecular Devices, San Jose, CA), while the camera and LED were triggered through TTL output through a Digidata 1440A (Molecular Devices) with Clampex 10.3 software (Molecular Devices).

During the acquisition process, experimenters were blind to the genotype and mRFP expression pattern of the GCaMP3$^+$ animals in both Method one and Method 2. After acquisition, images were de-identified using a random number generator to blind the analysis. For Method 1, images were stitched together as TIFF series in NIH ImageJ/FIJI (https://fiji.sc/). Due to the lack of paralysis in

Method 1, some contractions resulted in axial motion that temporarily removed the cells from the focal plane. These frames with cells outside of the focal plane were manually removed from the image series. For Method 2, image series were filtered to 2.5 Hz and exported from Metamorph in TIFF format. The Template Matching plugin (https://sites.google.com/site/qingzongtseng/template-matching-ij-plugin#description) for NIH ImageJ/FIJI was used to adjust for lateral motion during contractions and drift.

After alignment, regions of interest (ROIs) were drawn around each cell using the magic wand tool and the background (an area in each animal devoid of GCaMP3$^+$ cells) using the rectangle tool in NIH ImageJ/FIJI. The average gray values of these ROIs were measured over the time series using the multi measure tool in NIH ImageJ/FIJI. Raw data were exported to Excel (Microsoft, Redmond, WA) and fluorescence time series were converted using background subtraction to $\Delta F/F_0$ ($\Delta F/F_0 = (F - F_0)/F_0$), where $F_0$ was the baseline fluorescence for each trial. Kinetic measurements for individual peaks (rise time (10%–90%), decay time (90%–50%), and peak-width at half max) were made on the data acquired with Method two using Clampfit 10.7 (Molecular Devices).

Due to the mosaic nature of injections, each field contained 1–8 (median = 2, 25% quartile = 2, 75% quartile = 5) GCaMP3$^+$ myocytes. Further, individual cells exhibited 0–9 (median = 1, 25% quartile = 0, 75% quartile = 2.75) calcium transients within the imaging window. For $\Delta F/F_0$ quantitation, a unique $F_0$ was determined for each $Ca^{2+}$ transient event. In the case where a cell exhibited multiple $Ca^{2+}$ transient events, these events were treated as technical replicates and were averaged to give a single peak $\Delta F/F_0$ for each cell. In the case where a field contained multiple cells, the peak $\Delta F/F_0$ values for each cell were treated as technical replicates and averaged to give an average response for that animal. Cells or animals were considered biological replicates for analyses of peak $\Delta F/F_0$ and number of responses. For kinetic measurements, only the first peak in from each cell was analyzed and each peak was considered a biological replicate. For cells with only '-$\Delta F/F_0$' (photo-bleaching) recorded over the course of the trial, '0' was denoted as the peak $\Delta F/F_0$. Otherwise, the maximum numerical value of $\Delta F/F_0$ for each transient was assigned as peak $\Delta F/F_0$. For analysis of transient numbers, any transient with $\Delta F/F_0 \geq 0.05$ was counted as a response.

PCR genotyping was used to determine *ryr1b* alleles and assign data to its respective group. Only *ryr1b$^{+/+}$* and *ryr1b$^{mn0348Gt/mn0348Gt}$* animals were included in analyses due to the variable mRNA expression seen in *ryr1b$^{+/mn0348Gt}$* animals (*Clark et al., 2011a*).

## Forward genetic screening with next-generation sequencing

Isolated genomic DNA (300–500 ng) was digested with MseI, and BfaI in parallel for 3 hr at 37°C and heat inactivated for 10 min at 80°C. The digested samples from each enzyme were pooled with pre-aliquoted barcoded linker in individual wells. The T4 DNA ligase (M0202S, New England Biolabs, Inc, Ipswich, MA) was added, and the reaction mix was incubated for 2 hr at 16°C. The linker-mediated PCR was performed in two steps. In the first step, PCR was done with one primer specific to the 3'- ITR (R-ITR P1, 5'- AATTTTCCCTAAGTACTTGTACTTTCACTTGAGTAA-3') and the other primer specific to linker sequences (LP1, 5'- GTAATACGACTCACTATAGGGCACGCGTG- 3') using the following conditions: 2 min at 95°C, 25 cycles of 15 s at 95°C, 30 s at 55°C and 30 s at 72°C. The PCR products were diluted to 1:50 in dH2O, and a second round of PCR was performed using ITR (R-ITR P2, 5'-TCACTTGAGTAAAATTTTTGAGTACTTTTTACACCTC-3') and linker specific (LP2, 5' - GCGTGGTCGACTGCGCAT-3') nested primers to increase sensitivity and avoid non- specific amplification using the following conditions: 2 min at 95°C, 20 cycles of 15 s at 95°C, 30 s at 58°C and 30 s at 72°C. The nested PCR products from each 96-well plate are pooled and processed for Illumina library preparation as per manufacturer's instructions.

## Quantitative reverse transcription–PCR

To quantify knockdown efficiencies for a GBT-confirmed lines, GBT0235 carrying the RP2.1 insertion into the *lrpprc* gene locus, quantitative reverse transcription-PCR (qRT-PCR) was performed by using the following protocol. Embryo collections were obtained from in-crossed *lrpprc$^{+/mn0235Gt}$* adults. The larvae (six dpf) were sorted by mRFP expression to separate them based upon GBT allele and visible dark liver phenotype which has been characterized as a specific abnormality in *lrpprc$^{mn0235Gt/mn0235Gt}$* previously. Larvae with both RFP expression and dark liver phenotype were used as the experimental group (*lrpprc$^{mn0235Gt/mn0235Gt}$*) against larvae without either the RFP expression or

the liver phenotype as a control (*lrpprc* $^{+/+}$). After the initial sorting, individual zebrafish larvae were placed into a 1.7 ml tube with 350 µl RLT buffer within RNeasy Micro Kit (74004, QIAGEN) with β-mercaptoethanol (M6250, Sigma-Aldrich). Embryos were homogenized at max frequency (30 shakes/s) for 5 min using ~30 of 0.5 mm stainless steel beads, RNase free (SSB05, Next Advance, Troy, NY) and Tissue Lyser II (QIAGEN). Homogenized samples were replaced to MaXtract High Density tubes (129056, QIAGEN) to separate between the organic solvent (phenol/chloroform) and the nucleic acid-containing aqueous. Total RNA was purified from the nucleic acid-containing aqueous using the RNeasy Micro Kit (QIAGEN). 250 ng of total RNA from the individual larva was used for cDNA synthesis with the SuperScript II Reverse Transcriptase (18064014, Thermo Fisher Scientific) using random hexamer primers. The 16-folds diluted cDNA with deionized water were used as templates and no reverse transcriptase (RT) controls were run parallel to test for genomic DNA contamination. To analyze transcript levels, quantitative PCR was performed using SensiFAST SYBR Lo-ROX kit (BIO-94005, Bioline) and the CFX96 Touch Real-Time PCR Detection System (Bio-Rad, Hercules, CA). *eef1a1l1* levels were used as reference. Three technical replicates were run for each sample and three biological replicates for each group and two no RT controls were also run. Data were analyzed through calculation of Delta Ct values. Quantitative reverse transcription-PCR was repeated for four individual clutches from the pair of *lrpprc*$^{+/mn0235Gt}$ adults. Primer sequences are the following information; *lrpprc* FP: 5'-TGATAATGCTGAGGAAGCTCTCAAACTG-3', *lrpprc* RP: 5'-CCTTCATCTCCTTCAGTATGTCTAACGC-3', *eef1a1l1* FP: 5'-CCGTCTGCCAACTTCAGGATGTGT-3', *eef1a1l1* RP: 5'-TTGAGGACACCAGTCTCCAACACGA-3'. Source data can be found in *Figure 3—source data 1*.

## Annotating human orthologues of GBT-tagged genes and disease-causing genes

The human orthologues of 177 cloned zebrafish genes were mainly collected from Zebrafish Information Network (ZFIN, University of Oregon, Eugene, OR 97403–5274; URL: http://zfin.org/) In some cases, the candidates of human orthologues unlisted in ZFIN database were manually searched by using both Ensembl released 98 (https://useast.ensembl.org/index.html, (*Zerbino et al., 2018*) and InParanoid8 (http://inparanoid.sbc.su.se/cgi-bin/index.cgi, (*Sonnhammer and Östlund, 2015*) databases. In parallel, the candidates were manually identified by the result of Protein BLAST (https://blast.ncbi.nlm.nih.gov/Blast.cgi?PAGE=Proteins) assembled with human proteins and by the result of an online synteny analysis tool, SynFind in Comparative Genomics (CoGe) database (https://genomevolution.org/CoGe/SynFind.pl) (*Lyons and Freeling, 2008*). If the candidate multiply hit in those manual assessments, it was annotated as a human orthologue. The human phenotype data caused by mutations of 64 human orthologues were collected by using another data mining tool, BioMart provided by Ensembl (http://useast.ensembl.org/biomart/martview/) (*Kinsella et al., 2011*; *Smedley et al., 2015*).

## Protein classification of the trapped human orthologs

The 177 human orthologs of cloned GBT-tagged genes were analyzed using PANTHER14.1 (http://www.pantherdb.org/) (*Mi et al., 2019*). With those gene symbols, 176 gene were identified in the PANTHER system (the exception was *NRXN* Entrez Gene ID: 9378), and 105 genes were classified at least one PANTHER protein class (details are listed in *Figure 6—source data 1*). 8282/20996 human genes have been annotated with 214 protein classes in PANTHER14.1 (April, 2018, http://www.pantherdb.org/panther/summaryStats.jsp).

## Subcellular localization of the trapped human orthologs

Experimentally validated subcellular localization data of 177 human orthologous genes tagged by GBT were manually collected from the Human Protein Atlas Subcellular Localization data downloaded on August 27th, 2019 (Courtesy of Human Protein Atlas, www.proteinatlas.org) (*Uhlén et al., 2015*) and knowledge-based subcellular localization data for the 49 genes un-validated in the Human Protein Atlas was acquired from UniProtKB on Oct 2nd, 2019 (https://www.uniprot.org/) (*UniProt Consortium, 2018*). This data (*Supplementary file 4*) contains 271 entries because some human orthologous genes were annotated to multiple subcellular localizations.

### Finding disease models in vertebrates

Mouse models were found by using descriptions of animal models in both Online Mendelian Inheritance in Man (OMIM. Johns Hopkins University, Baltimore, MD: October 28th, 2019. URL: https://omim.org/) (*Amberger et al., 2019*) and in Mouse Genome Database (MGD) at the Mouse Genome Informatics (MGI) website, The Jackson Laboratory, Bar Harbor, Maine: October 28th, 2019 (URL: http://www.informatics.jax.org)(*Bult et al., 2019*). MGI provided the details of mouse models of human disease, such as the number of models that have been established. Zebrafish model were also found by using OMIM and ZFIN; August 28, 2019. ZFIN provided all data of fish strains listed in this database (*Ruzicka et al., 2019*). The area proportional Venn diagram were created using Bio-Venn (*Hulsen et al., 2008*) to visualize the number of human orthologs of the cloned genes associated with human genetic disorders which have at least one established disease model in zebrafish or mouse (*Figure 5B*).

### Gene expression profiling of the cloned zebrafish genes

The cloned genes with published expression data were isolated by using the wild-type expression data retrieved from ZFIN; August 28, 2019 (*Ruzicka et al., 2019*). In parallel, some published expression data were also manually searched from ZFIN. The mRFP reporter expression patterns of the cloned genes 2 and 4 dpf were manually searched using zfishbook database (*Clark et al., 2012*). The comparison with the number of genes with description about expression in both ZFIN and zfishbook (https://zfishbook.org/) was presented using BioVenn (*Hulsen et al., 2008*) to create the area proportional Venn diagrams in *Figure 7S* and *Figure 7T*.

### Statistical analysis

Knockdown efficiency and calcium imaging graphs were made in JMP 14 (SAS, Cary, NC) and in GraphPad Prism 8 (GraphPad Software, San Diego, CA), respectively. All other statistical analyses were performed with R (www.R-project.org) using R-Studio (www.rstudio.com/). Code used for analysis in R can be found in *Supplementary file 6*. Sample sizes for calcium imaging studies were estimated using peak $\Delta F/F_0$ data from Method one and the 'pwr' package (https://CRAN.R-project.org/package=pwr). Statistical analyses for calcium imaging data were performed using the 'wilcox.test' function or the 'coin' package (*Hothorn et al., 2006*; *Hothorn et al., 2008*) due to the non-normality visualized in the data. Outliers (determined by Grubb's test for one outlier in the 'outliers' package (https://CRAN.R-project.org/package=outliers) were included in overall analysis, although each statistical analysis was also performed without them as a proxy for the sensitivity of our conclusions to the outliers. All statistical tests supported the same conclusion with and without the outliers. Therefore, plots and p-values in the figures include all data points. p-values calculated excluding outliers can be found in *Supplementary file 6*. Effect size was measured by Cohen's d using the 'effsize' package (https://CRAN.R-project.org/package=effsize). For ease of reporting, all p-values less than 0.0001 were reported in figures as 'p<0.0001', but exact p-values are reported in *Supplementary file 6*.

### Availability of the materials and resources

All reagents are available upon request and all protein trap vectors in each reading frame will be deposited to Addgene (http://www.addgene.org). Zebrafish lines are available either from Zebrafish International Resource Center (ZIRC, http://zebrafish.org/) or the Mayo Clinic Zebrafish Facility, respectively.

## Acknowledgements

We thank Zoltan Varga for sharing of the ZIRC sperm cryopreservation protocol prior to publication, Vincent Cunliffe for sharing *p-mylpfa*:GCaMP3 plasmid, Sara Whiteman, Arthur Beyder, and Constanza Alcaino (Enteric Neuroscience Program, Mayo Clinic) for allowing us to use their $Ca^{2+}$ imaging setup and to Krista Habing and David Linden (Enteric Neuroscience Program, Mayo Clinic) for technical help with $Ca^{2+}$ imaging analysis. Additional thanks to the Mayo Clinic Media Services for providing the image in *Figure 5A*. Appreciation is also extended to the Mayo Clinic Zebrafish Facility staff for their excellent support.

## Additional information

### Funding

| Funder | Grant reference number | Author |
| --- | --- | --- |
| National Institutes of Health | GM63904 | Stephen C Ekker |
| National Institutes of Health | DA14546 | Stephen C Ekker |
| National Institutes of Health | DK093399 | Stephen C Ekker |
| National Institutes of Health | HG006431 | Stephen C Ekker |
| The Mayo Foundation | Internal | Stephen C Ekker |
| Natural Sciences and Engineering Research Council of Canada | RGPIN 05389-14 | Xiao-Yan Wen |
| The intramural Reserch Program of the National Human Genome Research Institute, National Institutes of Health | 1ZIAHG000183 | Shawn M Burgess |
| The Roy J. Carver Charitable Trust | 07-2991 | Maura McGrail<br>Jeffrey J Essner |
| Council of Scientific and Industrial Research | MLP1801 | Sridhar Sivasubbu |

The funders had no role in study design, data collection and interpretation, or the decision to submit the work for publication.

### Author contributions

Noriko Ichino, Data curation, Formal analysis, Investigation, Visualization, Methodology, Writing - original draft, Writing - review and editing; MaKayla R Serres, Lauren E Tallant, Resources, Investigation, Methodology, Writing - original draft; Rhianna M Urban, Mark D Urban, Kimberly J Skuster, Camden L Daby, Ying Wang, Hsin-kai Liao, Suzan El-Rass, Yonghe Ding, Weibin Liu, Resources, Investigation; Anthony J Treichel, Resources, Formal analysis, Validation, Investigation, Visualization, Methodology, Writing - original draft, Writing - review and editing; Kyle J Schaefbauer, Mark D Wishman, Investigation, Writing - original draft; Gaurav K Varshney, Data curation, Investigation, Methodology, Writing - original draft, Writing - review and editing; Melissa S McNulty, Resources, Methodology; Jennifer L Anderson, Investigation, Writing - original draft, Writing - review and editing; Ankit Sabharwal, Investigation; Lisa A Schimmenti, Data curation, Supervision, Writing - review and editing; Sridhar Sivasubbu, Supervision, Funding acquisition, Investigation; Darius Balciunas, Resources; Matthias Hammerschmidt, Steven Arthur Farber, Supervision, Investigation, Writing - review and editing; Xiao-Yan Wen, Maura McGrail, Jeffrey J Essner, Resources, Supervision, Funding acquisition, Investigation, Writing - review and editing; Xiaolei Xu, Resources, Supervision, Investigation, Writing - review and editing; Shawn M Burgess, Data curation, Supervision, Funding acquisition, Investigation, Methodology, Writing - review and editing; Karl J Clark, Conceptualization, Resources, Data curation, Supervision, Investigation, Visualization, Methodology, Writing - original draft, Project administration, Writing - review and editing; Stephen C Ekker, Conceptualization, Resources, Data curation, Supervision, Funding acquisition, Investigation, Visualization, Methodology, Writing - original draft, Project administration, Writing - review and editing

### Author ORCIDs

Noriko Ichino  https://orcid.org/0000-0002-7009-8299
Rhianna M Urban  https://orcid.org/0000-0001-6399-5015
Mark D Urban  https://orcid.org/0000-0002-9992-7820
Anthony J Treichel  http://orcid.org/0000-0002-4393-7034
Gaurav K Varshney  http://orcid.org/0000-0002-0429-1904
Suzan El-Rass  http://orcid.org/0000-0003-2075-4275
Ankit Sabharwal  https://orcid.org/0000-0003-4355-0355

Darius Balciunas [ID] http://orcid.org/0000-0003-1938-3243
Matthias Hammerschmidt [ID] http://orcid.org/0000-0002-3709-8166
Steven Arthur Farber [ID] http://orcid.org/0000-0002-8037-7312
Xiaolei Xu [ID] http://orcid.org/0000-0002-4928-3422
Maura McGrail [ID] http://orcid.org/0000-0001-9308-6189
Jeffrey J Essner [ID] http://orcid.org/0000-0001-8816-3848
Shawn M Burgess [ID] http://orcid.org/0000-0003-1147-0596
Karl J Clark [ID] https://orcid.org/0000-0002-9637-0967
Stephen C Ekker [ID] https://orcid.org/0000-0003-0726-4212

### Ethics

Animal experimentation: All zebrafish were maintained according to the guidelines and the standard procedures approved by the Mayo Clinic Institutional Animal Care and Use Committee (Mayo IACUC). The Mayo IACUC approved all protocols involving live vertebrate animals (A23107, A21710 and A34513).

### Decision letter and Author response

Decision letter https://doi.org/10.7554/eLife.54572.sa1
Author response https://doi.org/10.7554/eLife.54572.sa2

---

## Additional files

### Supplementary files

• Supplementary file 1. Genes disrupted in GBT-confirmed lines. Table lists the tagged genes (or unannotated coding sequence) of GBT-confirmed lines, genomic location and orientation of integrated loci, novel expression at 2 and 4 dpf, their human orthologs, and disease associations of their human orthologs. Blue text: published GBT-confirmed line, Red text: Integration locus in unannotated coding sequence, *: RNA sequencing reads mapping on the unannotated loci in at least one public dataset, †; zebrafish paralogs of GBT-tagged genes with one human ortholog, ‡: sequence of single 5' or 3' RACE product matched to two separate transcripts, *: integration locus mapped to GRCz11, $^{\gamma}$: line previously published as GBT0136, $^{d}$: denotes replicate genes with distinct integration events.

• Supplementary file 2. Homozygous phenotypes in GBT-confirmed lines. List of GBT-confirmed line number, tagged gene, a summary of their established phenotype, and references where more detailed characterization of each line can be found.

• Supplementary file 3. Publicly available human disease models. Established models of 32 human genetic disorders associated with 24 human orthologs of the GBT-tagged genes are generated by alternative genetic approaches in zebrafish and mice. This table listed the GBT ID of the tagged genes, both zebrafish tagged genes and their human orthologs, disease association of the human orthologs, disease ontology ID, number of models in zebrafish and mice listed in ZFIN (http://zfin.org/)(*Ruzicka et al., 2019*) and MGI (http://www.informatics.jax.org)(*Bult et al., 2019*) databases and references.

• Supplementary file 4. Subcellular localization of human orthologs tagged in GBT-confirmed lines. Subcellular localizations of 177 human orthologs tagged in GBT-confirmed lines were listed using experimental data from Human Protein Atlas (www.proteinatlas.org) (*Uhlén et al., 2015*) and knowledge base data from the UniProt knowledge base (UniProtKB, https://www.uniprot.org/, *UniProt Consortium, 2018*). *: UniProt annotation data, †: GO – Cellular Component, ‡: sequence of single 5' or 3' RACE product matched to two separate transcripts.

• Supplementary file 5. Oligo names and sequences.

• Supplementary file 6. The R code and output for sample size estimation and statistical analysis. Individual worksheets in *Figure 4—source data 1* and *Figure 4—figure supplement 1—source data 1* represent the individual '.csv' files read into R to perform these analyses and are named accordingly.

• Transparent reporting form

### Data availability

All data generated or analysed during this study are included in the manuscript and supporting files. Source data files have been provided for Figure 3, Figure 4, Figure 4-Figure Supplement 1, Figure 5 and Figure 6.

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
