## [Decision Letter]

**Decision letter after peer review:**

Thank you for sending your article entitled "Building the vertebrate codex using the gene breaking protein trap library" for peer review at *eLife* and very sorry for the delay in getting back to you. Your article is being evaluated by three peer reviewers, one of whom is a member of our Board of Reviewing Editors, and the evaluation is being overseen by a Reviewing Editor and Didier Stainier as the Senior Editor.

Given the list of essential revisions, including new experiments, the editors and reviewers invite you to respond with an action plan and timetable for the completion of the additional work. We plan to share your responses with the reviewers and then advise further with a formal decision. We fully understand that due to the current situation, it is difficult to predict how long the revisions will actually take, but we would still like to get an estimate of how much time you would need if the circumstances were normal.

Essential revisions:

1) No direct evidence for protein trapping is provided. The authors provided analysis of the subcellular localization of the protein product of GBT affected genes based on the PANTHER classification system. One of the promised powers of the GBT is the ability to show the subcellular localization of at least some of the affected proteins. Representative images of subcellular localization of mRFP that matches that of the endogenous proteins or PANTHER annotation will further demonstrate the utility of the resources.

2) The mutagenicity data are based on RP2 work that was published several years ago. The authors emphasize the development of RP8 in the manuscript, yet there are no data supporting that RP8 is as mutagenic as RP2. Although it is likely that they have similar mutagenicity, it is possible that the additional DNA may alter the mutagenicity.

3) The authors identified a GBT allele in each of 12 previously unannotated protein coding genes. While novel, the authors did not provide support for these claims with evidence such as expression in WT, expression patterns, gene structure, and integration sites. Only referencing the online materials is insufficient.

4) The manuscript reports 12 embryonic lethal mutations, yet only mentioned the gene name of 6 published mutants. Phenotypic description of the 6 other embryonic lethal mutations will make them a better resource for the community.

5) The manuscript reports 1200 GBT lines, but only 213 of them have insertion site mapped. Since unmapped lines are less interesting, the authors need to propose a way forward. Are the unmapped lines unmappable, or is there a way forward? If there is a way forward, why was this not already done?

6) In the Results section, it is difficult for the readers to distinguish published results from unpublished results. The authors need to make them clear.

*Reviewer #1:*

The manuscript reports a resource of more than 1,200 zebrafish mutant lines generated by a gene-trap approach. Some of the collection was generated by improved gene breaking transposons (GBT) that differ from previous published GBT from the group in two ways: 3 vectors that covers all 3 reading frames by the 5' trap and a lens-specific promoter-driven BFP for the 3' trap. One feature of these mutant lines is that each allele carries a mRFP tag allowing live imaging of the expression pattern of the affected gene. The authors have acquired dorsal, ventral, and sagittal images of mRFP signal at 2 and 4 dpf for each allele and uncovered many previous undescribed expression patterns. The images can be accessed on a previously described website. The authors have also cryopreserved each line and deposited one copy of the library at the Zebrafish International Resource Center for distribution. Another feature of these alleles is that they are potentially revertible by Cre. Despite the ease of generating mutations in zebrafish nowadays, these features may still make the collection a useful resource for zebrafish investigators. In addition, these alleles cause >97% reduction of the mRNA and recapitulate published phenotypes in mutant generated using other approaches. Many of the alleles affect genes whose human orthologs have been implicated in diseases. The mutant collection therefore may be a useful resource for human disease modeling and mechanism studies. As an example, the authors characterized muscle Ca^2+^ transients in the ryr1b homozygous mutant and demonstrated substantial decrease of the amplitude and increase of rising time of spike, consistent with its known function in regulating sarcoplasmic reticulum Ca^2+^ release. In addition, the authors identified a number of novel expressed loci. Overall, the manuscript describes an unique resource important for the zebrafish field and have the potential to accelerate discovery and disease modeling. However, there are still a number of issues that needs to be addressed for making the resources more attractive to other investigators.

Essential revisions:

1) One of the claimed advantages of insertional mutagenesis approach is the ease of identification of the affected loci. Knowing the affected gene in a line will tremendously increase its usage. Of the 1200 lines, however, only 213 have the integration site and affected gene determined. These include previously published 40 lines. Yet the authors touted the advantage of their NGS-based cloning pipeline. The reason for the gap is not clearly stated, although the authors mentioned poor genome annotation.

2) In the Results section, the distinction between published results and new results is not clearly. For example, it seems that the mutagenic efficiency of GBT section is almost entirely based on published data except for irpprc and eef1al1. The same seems to be true for the phenotype appearance rate. The authors should make state clearly what results are new in the manuscript.

3) The authors claim that mRFP expression enabled them to identify new expression patterns of the trapped gene. At least for expression at 2 dpf, it is not clear if the new expression patterns are due to ectopic expression of mRFP, or incomplete annotation of previous results. They author should provide additional supporting evidence, such as in situ hybridization results, to validate that mRFP recapitulates endogenous mRNA expression patterns.

4) The authors claimed that they identified 13 novel genes but did not provide additional supporting evidence other than the detection of fusion transcripts. More information, such as additional expression evidence of these genes and the existence of their orthologs in human or other species would provide more confidence.

*Reviewer #2:*

Nine out of every 10 human genes has yet to be functionally characterized. To address this issue, Ichino et al. designed and tested several generations of gene trap constructs for probing expression pattern and gene function in the powerful vertebrate model system, the zebrafish. The work presented explains the design of an extremely clever three reading frame, fluorescently tagged and experimentally revertible splice acceptor-based gene trap vector, which they call the gene-break transposon protein trap ("GBT system"). The injection of vector and transposase into zebrafish eggs yielded some 1200 insertion lines in a powerful model system, the zebrafish. Heterozygous insertions yield normal expression pattern through larval development (and potentially beyond); homozygosing these mutagenizing insertions yields mutant fish whose phenotypes yield functional insights. This work lays a powerful foundation for potentially studying the expression patterns and functions of every gene in the vertebrate genome. The results include the discovery of previously unknown expression patterns for 91% of their trapped genes, based on imaging at 4 days post-fertilization. Mutant lines were shown to phenocopy existing mutants, and include models of human genetic disease. Overall, they authors present a compelling basis for consideration of the "vertebrate codex" – a term I find compelling.

Introduction: The Abstract was well-written, but the last sentence of the Introduction is overstated. Since phenotyping is dependent upon assays that may or may not have been used, it would seem more accurate to say something like, "…this GBT system lays a foundation for functional annotation of the vertebrate genome…" It would also add clarity to community thinking to explicitly note that mutant phenotyping is a key to functional annotation of genes.

Subsection “GBT protein trapping generates a variety of potential models of human disease”: The consideration of RYR1 in discussion would better be placed in one, not two, sections. The degree of repetition should be reduced.

Subsection “Gene-break transposon system as a next generation mutagenesis system”: The limitations of the old gene trap lines were already explained in the Results section. Emphasis in the Discussion section should focus on benefits of the new system.

Subsection “Gene-break transposon system as a next generation mutagenesis system”: The second paragraph seems to repeat what is in the Results section.

Figure 4: There is a striking degree of overlap in the beeswarm plots. It would be helpful to make the utility of these results more clear or to perhaps detail only the key aspects of the model, enough to be convincing of the focus of the paper – the novelty and utility of the gene-break transposon approach.

*Reviewer #3:*

The authors published a paper describing a protein trap screen in 2011 in which they collected 350 patterns and characterized 40 lines extensively (Table 1 in the 2011 paper). The current manuscript is the expansion of the previous work. In this manuscript, (1) they constructed new protein trap vectors, (2) they collected 1200 lines (patterns) and performed molecular characterization of 213 lines (GBT-confirmed lines), (3) based on the trapped gene information, they grouped the trap lines from views of human diseases, ontology, and subcellular localization, (4) calcium imaging using the ryr1b mutant. This should have been a lot of work and the resources will be useful to the community. However, the advances made since work published in 2011 are unclear.

1) They constructed RP2 series vectors that contained RFP gene in three frames. Also, the RP8 series vectors have RFP in three frames and the crystalline promoter. I admit this is an interesting idea. However when one of them was used for injection, introns which cannot be targeted by the other vectors may be targeted, but at the same time it cannot target introns which can be targeted by the other vectors. Thus, a set of new vectors may increase coverage of the genome, but I do not think the modifications affect effectiveness of the mutagenesis screen. Also, the number of insertions created by using each vector is too small to evaluate the vectors and their idea. They need to mention such limitations.

2) In Supplementary file 1, most (nearly all) of the lines were created by using RP2.1. Since the authors reported the other new vectors, they need to evaluate those vectors with actual data (trapping events and efficiencies, etc). They need to show the power of these vectors.

3) Their indication of "97% knockdown efficiency" was already described in the 2011 paper. In this manuscript, since they constructed new RP8 series, they should measure the efficiencies of these vectors and discuss in comparison with the RP2 series.

4) In this manuscript, they described the nature of the traps mainly by based on the gene information (in relation to human disease genes, gene ontology (PANTHER), and subcellular localization). The power of this system is visualization of subcellular localization of the trapped protein. The expression patterns can be visualized without the protein trap mechanism. They need to show the data for subcellular localization of a certain number of the trapped proteins (at least more than 10?). They should show co-localization of the fusion protein and endogenous protein if the antibody is available.

5) ryr1b gene and mutants had been extensively reported by Hirata et al., (2007) and by themselves (Clark et al., 2011). What is new here is calcium imaging of muscle activities in the ryr1b mutant. However, the same result was obtained and described in Hirata et al., (2007) by electrophysiology. Thus, the novelty is poor. They should describe analysis of another gene which was not described previously.

6) They identified 12 homozygous lethal mutants, but I do not see any data for this. The authors should show the data. 5 mutants they mentioned in the text were already described in the previous manuscript.

7) In Table 3, they say they identified 12 new previously-unannotated genes. What are those genes? and how they were trapped by the vector? More data should be shown.

---

## [Author Response]

Essential revisions:1) No direct evidence for protein trapping is provided. The authors provided analysis of the subcellular localization of the protein product of GBT affected genes based on the PANTHER classification system. One of the promised powers of the GBT is the ability to show the subcellular localization of at least some of the affected proteins. Representative images of subcellular localization of mRFP that matches that of the endogenous proteins or PANTHER annotation will further demonstrate the utility of the resources.

We certainly appreciate the concern about subcellular localization data gained through GBT protein trapping. In our previous work, the subcellular localizations have, in most cases, been consistent with what is expected. These observations lead us to perform the new analyses based upon PANTHER and Human Protein Atlas instead using a high resolution imaging approach. We realize that we did not explicitly state this or cite these previous publications in this section. In the revision, we have added citations to our previous work and our rational for these analyses.

Similarly, we agree that showing some new protein trapping expression data, especially some that we can relate to our PANTHER and/or Human Protein Atlas analyses, is useful for readers. We have therefore included confocal images of the GBT-confirmed lines GBT0235 (*lrpprc*) and GBT0348 (*ryr1b*) to highlight distinctive subcellular localization patterns at the top of the revised Figure 6. The human ortholog of *lrpprc* maps to mitochondria in Human Protein Atlas, but fails to map to a PANTHER class. The human ortholog of *ryr1b* maps to the cytosol, Golgi, and vesicles in Human Protein Atlas and is classified as a transporter in PANTHER analysis. We have also added additional confocal images that show distinct subcellular localizations in GBT-candidate lines and GBT lines to further demonstrate protein trapping in these lines in Figure 6—figure supplement 1.

2) The mutagenicity data are based on RP2 work that was published several years ago. The authors emphasize the development of RP8 in the manuscript, yet there are no data supporting that RP8 is as mutagenic as RP2. Although it is likely that they have similar mutagenicity, it is possible that the additional DNA may alter the mutagenicity.

We acknowledge that the RP2 mutagenicity data is an aggregate of prior work plus GBT0235 (*lrpprc*). However, given the regular questions we have received on this innovative mutagen, we took the opportunity to conduct and present this novel analysis compiling our mutagenicity data and the comparison to other protein trapping methods. We also describe here the first description of the two additional RP2-based vectors with only a single nucleotide difference from RP2.1 to enable the capture in any protein reading frame.

The RP8 variants were built for two reasons – to avoid the use of GFP as a reporter and to be a modular molecular cassette for others to use (i.e. with non-transposon genetic engineering methods such as GeneWeld targeted integration, (Wierson et al., 2018)). We report that the protein trap component is functioning as expected (and 18 of the GBT-confirmed lines we present contain RP8 integrations). We also included the same 1.6kb poly(A)/presumptive scaffold attachment transcriptional termination cassette we showed to be effective mutagens with the PX and RP2 vectors. Given that the major functional change to RP8 lies in its 3’ exon trap, we expect its mutagenicity to be similar to RP2, but we don’t have the data to confirm that it is the same at this time. We therefore have explicitly acknowledged in the revisions (subsection “Gene-break transposon system as a next generation mutagenesis system”) that we have not performed the analyses of RP8 mutagenicity and kept the RP8 vector description with this caveat.

3) The authors identified a GBT allele in each of 12 previously unannotated protein coding genes. While novel, the authors did not provide support for these claims with evidence such as expression in WT, expression patterns, gene structure, and integration sites. Only referencing the online materials is insufficient.

We appreciate the interest in the previously unannotated transcripts. We have spent additional time analyzing these loci using additional publicly available resources, our own in-house RNA-Seq datasets and GRCz10. We focused on WT embryos at 4 different developmental stages, 36 hpf, 48 hpf, and 4 dpf (Data source: ftp://ftp.ensembl.org/pub/data_files/danio_rerio/ across). The detail of this analysis is described in the revised method section. At the end of this additional work, we were able to resolved 7 lines (GBT0148, GBT0264, GBT0724, GBT1024, GBT1100 and GBT1168) into new cloned loci with likely transcripts (in the revised Supplementary file 1). In total, we resolved 7 out of these 10 unannotated GBT lines (in the revision) and moved this into our main catalog. Since we have informed genomic location of all 204 integration loci in the revised Supplementary file 1, we removed the previous Table 3 with the redundancy. We have also removed the individual section discussing Table 3. Instead, we have added the following text:

In subsection “Molecular cloning of a subset of GBT lines highlights the genetic diversity of this protein trap collection”, “A small subset of these GBT-confirmed lines mapped to areas in the genome without annotated transcripts. Publicly available RNA-sequencing data revealed reads flanking a majority of these mapped integrations. Some of these reads contained evidence of splicing in the sense orientation of the mRFP reporter (Supplementary file 1)”.

In subsection “GBT protein trapping provides the basis for annotation of functionally diverse proteins and novel transcripts mapped on poorly assembled genomic regions” it now reads, “We indeed found that 10 population of GBT integrations in the confirmed lines (with mRFP expression) failed to map to any predicted gene. However, RNA sequencing reads in public datasets identified potential unannotated coding sequences aligned with these GBT integration loci (Supplementary file 1). While 5' and 3' RACE are necessary to confirm the mRNA fusion products, these unannotated coding sequences represent the possibility to annotate novel, protein-coding transcripts in these GBT lines. Therefore, GBT protein trapping can find, illuminate expression, and elucidate in vivo functions of novel genes and/or gene variants in poorly annotated regions of reference genomes”.

4) The manuscript reports 12 embryonic lethal mutations, yet only mentioned the gene name of 6 published mutants. Phenotypic description of the 6 other embryonic lethal mutations will make them a better resource for the community.

Phenotypic descriptions definitely make a line particularly valuable for the community. A number of these lines with embryonic phenotypes have been cloned and published following this initial forward genetic screening assessment. Further, additional non-embryonic phenotypes have been discovered in some GBT-confirmed lines. Overall, 17 of our GBT-confirmed lines have been published or characterized to have a phenotype, ranging from embryonic lethal to reduced adult viability to differences in drug susceptibility. We have therefore created a new table (Supplementary file 2 in the revision) that contains the GBT-confirmed line number, zebrafish gene symbol, a description of the phenotype observed, and a reference to the publication where that has been described to make this information readily available to the readers. We added the following text in the Phenotype appearance in GBT lines is comparable to other mutagenic technologies for forward genetic screening section of the results: “To date, 17 of our GBT-confirmed lines have been published with homozygous phenotypes ranging from embryonic lethal, to reduced adult viability, to differences in pharmacological susceptibility (Supplementary file 2). Additional GBT-confirmed lines with homozygous phenotypes will continue to be identified and characterized”.

5) The manuscript reports 1200 GBT lines, but only 213 of them have insertion site mapped.

Mapping confirmation is a technical throughput bottleneck in our work due to its multi-step and the artisanal nature of this process: (1) Initial isolation of a candidate genomic sequence (this was conducted in a parallel molecular screening process). (2) Manual design of primers to this genomic sequence and linkage confirmation to mRFP against cDNA generated from mRFP-positive embryos/larvae.

We understand that ~200 out of 1200 lines may have been portrayed as a small contribution, but every single of the 147 new lines reported here were all manually confirmed using this time-intensive process. We take this comment as an opportunity to elaborate on the number of new GBT-confirmed lines that this manuscript offers the community. We thus added text to emphasize the following points. (1) Before this manuscript, we and our collaborators had published 57 total GBT-confirmed lines across five studies. (2) The present study alone provides an additional 147 unique GBT-confirmed lines, including 15 GBT lines that had been used in previous studies without their trapped genes being mapped. Many of these new GBT-confirmed lines stemmed from successful confirmation of GBT-candidate lines mapped with our next-generation sequencing pipeline.

Since unmapped lines are less interesting, the authors need to propose a way forward. Are the unmapped lines unmappable, or is there a way forward? If there is a way forward, why was this not already done?

The next-generation sequencing pipeline revealed candidate genes for an additional 144 GBT lines in the revision. These lines have simply not undergone the comprehensive manual confirmation/cloning process but have a full way forward to molecular isolation (as do all of the other hundreds of lines with only a described expression pattern). Indeed, between methods like inverse/splinkerette PCR and 5’ and 3’ RACE, we are confident nearly all protein trap lines can be readily mapped and confirmed. We have added an explicit section with the following description, “Gene-break protein trap library is a rich resource for the community” in the discussion on the road forward for (1) 144 candidate lines and (2) over 800 GBT lines with a novel expression pattern for any member of the zebrafish community. “Taken together, GBT-based mRFP-reporters demonstrate how much we still have left to understand about the expression patterns of the overall proteome and, ultimately, the complex codex that is our genome. Even at the relatively well-studied 2-dpf stage, nearly 40% of GBT-confirmed lines elucidated novel gene expression data (Figure 7). Cataloging these expression patterns enables investigators to make collections of lines with expression in their cell/tissue of interest and/or a phenotype. The remaining 144 GBT-candidate lines and over 800 GBT lines represent a rich resource for genomic discoveries. For any GBT-candidate or GBT lines of interest, a similar cloning pipeline (Figure 2) can be employed to identify the GBT integration locus. In addition, the refinement of the zebrafish genome will enhance our ability to complete the annotation from GBT-line to GBT-confirmed line for any given line with a desired expression profile and/or phenotype. Together, this 1,200+ GBT-line collection is a new contribution for using zebrafish to annotate the vertebrate genome.”

6) In the Results section, it is difficult for the readers to distinguish published results from unpublished results. The authors need to make them clear.

We thank the editor/reviewers for this comment! We realize in our compendium mindset and the nature of how we wrote this paper as an overview that we made it very difficult to distinguish published data from novel results. We appreciate the opportunity to focus on highlighting the new data that we present. We have therefore prioritized clear separation of published and unpublished results. We have taken the following actions in a revised manuscript to better assist readers in finding new data in this manuscript:

1) At the beginning of the third paragraph of the Introduction, we changed the wording from “with a single GBT protein trap construct” to “In the original GBT protein trap construct” to signal to previous work that we have published. We also denoted RP2.1 as the “original” GBT protein trap construct in the remainder of the paper.

2) In the middle of the third paragraph of the Introduction, we modified the sentence to include the word “new” when referring to the constructs generated in this paper. In the same sentence, we changed from stating “over 1200” to “over 800 additional” to acknowledge the cumulative nature of the project while highlighting the new numbers in this paper as follows: “Alongside the original, we employed these new vectors in zebrafish to generate and catalog over 800 additional GBT protein trap lines with visible mRFP expression at 2 dpf (end of embryogenesis) or 4 dpf (larval stage)”. We also employ “over 800 additional” instead of “over 1200” when first describing the creation of the collection of the GBT lines in this manuscript.

3) In the last half of the third paragraph of the Introduction, we only highlight new analyses and data presented in this manuscript and remove any reference to data that has been previously published to give readers and overview to focus on the novel work in this paper.

4) We added a key citation to (Clark et al., 2011) in the middle of the second paragraph of subsection “GBT vector series RP2 and RP8 illuminate all three vertebrate proteomic reading frames” that provides the motivation to develop GBT constructs that trap expression in all three reading frames and to develop RP8 with a new 3’ exon trap.

5) We revised the end of the second paragraph of subsection “GBT vector series RP2 and RP8 to illuminate all three vertebrate proteomic reading frames” to be consistent with Figure 1—figure supplement 1 and only demonstrate the new expression off of RP2.2, RP2.3 and the RP8 series as follows: “Using all five of these new GBT constructs in zebrafish, we conducted an initial screen for expression of protein trap mRFP and observed that all RP2 and RP8 series constructs readily produced mRFP fusion proteins expressed from their endogenous promoters (Figure 1—figure supplement 1)”.

6) We specified how many of the GBT-confirmed lines in the 200+ collection are newly published as confirmed lines in this manuscript at the beginning of the subsection “Molecular cloning of a subset of GBT lines highlights the genetic diversity of this protein trap collection” as follows: “147 of these GBT-confirmed lines are newly characterized in this manuscript and were selected for confirmation based upon their expression pattern and/or homozygous phenotype (Supplementary file 1)”.

7) Under subsection “RP2.1 induces high knockdown efficiency of endogenous transcripts in GBT-confirmed lines” , we clarified that this section is the result of data compilation from previous publications with one additional dataset in the RP2.1 group as follows: “We wanted to know the knockdown efficacy of the GBT system as a quantitative assessment of mutagenicity. We therefore compiled qRT-PCR data to compare wild-type and truncated, mRFP-fused transcript levels for all 26 RP2.1-derived GBT-confirmed lines that we and others have tested (Clark et al., 2011; Ding et al., 2013; Ding et al., 2016), GBT0235 (RP2.1)—this manuscript). This compilation determined at minimum 97% knockdown in animals homozygous for the RP2.1 alleles”.

8) We now point out data that has been published before that is being reported in the Results section using words such as “published”, “compiled”, “initial”, “original”, “previously”, “reported”, and “previous study”.

9) Where unpublished data is juxtaposed with published data, we explicitly point out the unpublished data using words such as “new”, “additional”, “novel”, “in this study”, and “novel approach”.

10) While Figure 5 and Figure 6 do include some previously published GBT-confirmed lines, the analysis of human disease-associated genes, PANTHER classification, and subcellular localization via Human Protein Atlas are all new for this manuscript.

11) Figure 7 also represents a compilation of the entire GBT-confirmed line collection. However, all images in this figure are newly described and the comparison between zfishbook and ZFIN is entirely novel.

Reviewer #1:The manuscript reports a resource of more than 1,200 zebrafish mutant lines generated by a gene-trap approach. Some of the collection was generated by improved gene breaking transposons (GBT) that differ from previous published GBT from the group in two ways: 3 vectors that covers all 3 reading frames by the 5' trap and a lens-specific promoter-driven BFP for the 3' trap. One feature of these mutant lines is that each allele carries a mRFP tag allowing live imaging of the expression pattern of the affected gene. The authors have acquired dorsal, ventral, and sagittal images of mRFP signal at 2 and 4 dpf for each allele and uncovered many previous undescribed expression patterns. The images can be accessed on a previously described website. The authors have also cryopreserved each line and deposited one copy of the library at the Zebrafish International Resource Center for distribution. Another feature of these alleles is that they are potentially revertible by Cre. Despite the ease of generating mutations in zebrafish nowadays, these features may still make the collection a useful resource for zebrafish investigators. In addition, these alleles cause >97% reduction of the mRNA and recapitulate published phenotypes in mutant generated using other approaches. Many of the alleles affect genes whose human orthologs have been implicated in diseases. The mutant collection therefore may be a useful resource for human disease modeling and mechanism studies. As an example, the authors characterized muscle Ca^2+^ transients in the ryr1b homozygous mutant and demonstrated substantial decrease of the amplitude and increase of rising time of spike, consistent with its known function in regulating sarcoplasmic reticulum Ca^2+^ release. In addition, the authors identified a number of novel expressed loci. Overall, the manuscript describes an unique resource important for the zebrafish field and have the potential to accelerate discovery and disease modeling. However, there are still a number of issues that needs to be addressed for making the resources more attractive to other investigators.Essential revisions:1) One of the claimed advantages of insertional mutagenesis approach is the ease of identification of the affected loci. Knowing the affected gene in a line will tremendously increase its usage. Of the 1200 lines, however, only 213 have the integration site and affected gene determined. These include previously published 40 lines. Yet the authors touted the advantage of their NGS-based cloning pipeline. The reason for the gap is not clearly stated, although the authors mentioned poor genome annotation.

We have added context for these 200plus lines and a discussion on the way forward for the unmapped lines.

Mapping confirmation is a technical throughput bottleneck in our work due to its multi-step and the artisanal nature of this process: (1) Initial isolation of a candidate genomic sequence (this was conducted in a parallel molecular screening process). (2) Manual design of primers to this genomic sequence and linkage confirmation to mRFP against cDNA generated from mRFP-positive embryos/larvae.

We understand that ~200 out of 1200 lines may have been portrayed as a small contribution, but every single of the 147 new lines reported here were all manually confirmed using this time-intensive process. We take this comment as an opportunity to elaborate on the number of new GBT-confirmed lines that this manuscript offers the community. We thus added text to emphasize the following points. (1) Before this manuscript, we and our collaborators had published 57 total GBT-confirmed lines across five studies. (2) The present study alone provides an additional 147 unique GBT-confirmed lines, including 15 GBT lines that had been used in previous studies without their trapped genes being mapped. Many of these new GBT-confirmed lines stemmed from successful confirmation of GBT-candidate lines mapped with our next-generation sequencing pipeline.

2) In the Results section, the distinction between published results and new results is not clearly. For example, it seems that the mutagenic efficiency of GBT section is almost entirely based on published data except for irpprc and eef1al1. The same seems to be true for the phenotype appearance rate. The authors should make state clearly what results are new in the manuscript.

We acknowledge that the mutagenicity data, with the exception of *lrpprc*, was compiled from previous publications. We have added signal words to state which results represent new data and which rely on previously published information. Please see “Essential revision 2 and Essential revision 6”. In addition, we quantified the expression levels of *eef1al1* as a reference gene to normalize the expression level of *lrpprc* in both WT and *lrpprc* mutant animals. (Please see the Materials and methods section.)

3) The authors claim that mRFP expression enabled them to identify new expression patterns of the trapped gene. At least for expression at 2 dpf, it is not clear if the new expression patterns are due to ectopic expression of mRFP, or incomplete annotation of previous results. They author should provide additional supporting evidence, such as in situ hybridization results, to validate that mRFP recapitulates endogenous mRNA expression patterns.

We acknowledge that this point is ambiguous due to the nature of the word “description” in this section of the results that could mean additional (possibly ectopic) expression, incomplete annotation of previous images, or novel expression for genes without any known expression patterns. This analysis was performed to highlight genes that our GBT-confirmed lines provided the first publicly available expression patterns. We have therefore modified wording to explicitly say, “Our GBT-confirmed lines revealed expression patterns (available on www.zfishbook.org) for 67 genes at 2 dpf and 173 genes at 4 dpf without publicly available expression data in ZFIN”.

4) The authors claimed that they identified 13 novel genes but did not provide additional supporting evidence other than the detection of fusion transcripts. More information, such as additional expression evidence of these genes and the existence of their orthologs in human or other species would provide more confidence.

We appreciate the interest in the previously unannotated transcripts. We have spent additional time analyzing these loci using additional publicly available resources, our own in-house RNA-Seq datasets and GRCz10. We focused on WT embryos at 4 different developmental stages, 36 hpf, 48 hpf and 4 dpf (Data source: ftp://ftp.ensembl.org/pub/data_files/danio_rerio/ across). The detail of this analysis is described in the revised method section. At the end of this additional work, we were able to resolved 7 lines (GBT0148, GBT0264, GBT0724, GBT1024, GBT1100 and GBT1168) into new cloned loci with likely transcripts (in the revised Supplementary file 1). In total, we resolved 7 out of these 10 unannotated GBT lines (in the revision) and moved this into our main catalog. Since we have informed genomic location of all 204 integration loci in the revised Supplementary file 1, we removed the previous Table 3 with the redundancy. We have also removed the individual section discussing Table 3. Instead, we have added the following text:

In subsection “Molecular cloning of a subset of GBT lines highlights the genetic diversity of this protein trap collection”, “A small subset of these GBT-confirmed lines mapped to areas in the genome without annotated transcripts. Publicly available RNA-sequencing data revealed reads flanking a majority of these mapped integrations. Some of these reads contained evidence of splicing in the sense orientation of the mRFP reporter (Supplementary file 1)”.

In subsection “GBT protein trapping provides the basis for annotation of functionally diverse proteins and novel transcripts mapped on poorly assembled genomic regions” it now reads, “We indeed found that 10 population of GBT integrations in the confirmed lines (with mRFP expression) failed to map to any predicted gene. However, RNA sequencing reads in public datasets identified potential unannotated coding sequences aligned with these GBT integration loci (Supplementary file 1). While 5' and 3' RACE are necessary to confirm the mRNA fusion products, these unannotated coding sequences represent the possibility to annotate novel, protein-coding transcripts in these GBT lines. Therefore, GBT protein trapping can find, illuminate expression, and elucidate in vivo functions of novel genes and/or gene variants in poorly annotated regions of reference genomes”.

Reviewer #2:Nine out of every 10 human genes has yet to be functionally characterized. To address this issue, Ichino et al. designed and tested several generations of gene trap constructs for probing expression pattern and gene function in the powerful vertebrate model system, the zebrafish. The work presented explains the design of an extremely clever three reading frame, fluorescently tagged and experimentally revertible splice acceptor-based gene trap vector, which they call the gene-break transposon protein trap ("GBT system"). The injection of vector and transposase into zebrafish eggs yielded some 1200 insertion lines in a powerful model system, the zebrafish. Heterozygous insertions yield normal expression pattern through larval development (and potentially beyond); homozygosing these mutagenizing insertions yields mutant fish whose phenotypes yield functional insights. This work lays a powerful foundation for potentially studying the expression patterns and functions of every gene in the vertebrate genome. The results include the discovery of previously unknown expression patterns for 91% of their trapped genes, based on imaging at 4 days post-fertilization. Mutant lines were shown to phenocopy existing mutants, and include models of human genetic disease. Overall, they authors present a compelling basis for consideration of the "vertebrate codex" – a term I find compelling.Introduction: The Abstract was well-written, but the last sentence of the Introduction is overstated. Since phenotyping is dependent upon assays that may or may not have been used, it would seem more accurate to say something like, "…this GBT system lays a foundation for functional annotation of the vertebrate genome…" It would also add clarity to community thinking to explicitly note that mutant phenotyping is a key to functional annotation of genes.

We have modified the last sentence of the Introduction to state, “Since detailed investigations of mutant phenotypes are vital to functional annotation of the vertebrate genome, the mutagenic reporters in our GBT system provide the basis for this functional annotation to better understand normal biology and human disease”.

Subsection “GBT protein trapping generates a variety of potential models of human disease”: The consideration of RYR1 in the Discussion section would better be placed in one, not two, sections. The degree of repetition should be reduced.

We have condensed these two sections into one to link the consideration of our *ryr1b* results with *RYR1* mutations in humans and the implications this has for future disease modeling with GBT-confirmed lines.

Subsection “Gene-break transposon system as a next generation mutagenesis system”: The limitations of the old gene trap lines were already explained in the Results section. Emphasis in the Discussion section should focus on benefits of the new system.

We have modified subsection “Gene-break transposon system as a next generation mutagenesis system” to highlight the benefits of the new system and its applications toward future genome engineering approaches using the RP8 series.

Subsection “Gene-break transposon system as a next generation mutagenesis system”: The second paragraph seems to repeat what is in the Results section.

We have modified subsection “Gene-break transposon system as a next generation mutagenesis system” to be less repetitive with what is in the Results section.

Figure 4: There is a striking degree of overlap in the beeswarm plots. It would be helpful to make the utility of these results more clear or to perhaps detail only the key aspects of the model, enough to be convincing of the focus of the paper – the novelty and utility of the gene-break transposon approach.

We have further explained our basis for including the violin plots of both amplitude, peak-width at half-max, rise time, and decay time. We have also modified this section to focus on validating the GBT system as a way to functionally annotate the genome with a comparison of our GBT *ryr1b* allele to the previously published *relatively relaxed* allele.

Reviewer #3:The authors published a paper describing a protein trap screen in 2011 in which they collected 350 patterns and characterized 40 lines extensively (Table 1 in the 2011 paper). The current manuscript is the expansion of the previous work. In this manuscript, (1) they constructed new protein trap vectors, (2) they collected 1200 lines (patterns) and performed molecular characterization of 213 lines (GBT-confirmed lines), (3) based on the trapped gene information, they grouped the trap lines from views of human diseases, ontology, and subcellular localization, (4) calcium imaging using the ryr1b mutant. This should have been a lot of work and the resources will be useful to the community. However, the advances made since work published in 2011 are unclear.1) They constructed RP2 series vectors that contained RFP gene in three frames. Also, the RP8 series vectors have RFP in three frames and the crystalline promoter. I admit this is an interesting idea. However when one of them was used for injection, introns which cannot be targeted by the other vectors may be targeted, but at the same time it cannot target introns which can be targeted by the other vectors. Thus, a set of new vectors may increase coverage of the genome, but I do not think the modifications affect effectiveness of the mutagenesis screen. Also, the number of insertions created by using each vector is too small to evaluate the vectors and their idea. They need to mention such limitations.

We acknowledge that a single vector cannot be a one size fits all solution for all introns. We have added a sentence in the Discussion section to explicitly point out that each vector can only target a subset of introns as follows, “While each individual construct still only integrates in-frame in a subset of introns, the RP2 series potentiates in-frame mRFP for any intron with Tol2-mediated integration”. We also explicitly highlight that having all three reading frames available offers the highest utility for future targeted integration approaches using RP8 as follows, “The three reading frames and modularity of the RP8 series are especially well suited to targeted integration approaches”.

2) In Supplementary file 1, most (nearly all) of the lines were created by using RP2.1. Since the authors reported the other new vectors, they need to evaluate those vectors with actual data (trapping events and efficiencies, etc). They need to show the power of these vectors.

We acknowledge that nearly all lines were made using RP2.1 and all of the mutagenicity data conducted using RP2.1. We also describe here the first description of the two additional RP2-based vectors with only a single nucleotide difference from RP2.1 to enable the capture in any protein reading frame. We show effective protein trapping with all RP2 series and RP8 series vectors.

3) Their indication of "97% knockdown efficiency" was already described in the 2011 paper. In this manuscript, since they constructed new RP8 series, they should measure the efficiencies of these vectors and discuss in comparison with the RP2 series.

The RP8 variants were built for two reasons – to avoid the use of GFP as a reporter and to be a modular molecular cassette for others to use (i.e. with non-transposon genetic engineering methods such as GeneWeld targeted integration, (Wierson et al., 2018)). We report that the protein trap component is functioning as expected (and 18 of the GBT-confirmed lines we present contain RP8 integrations). We also included the same 1.6kb poly(A)/presumptive scaffold attachment transcriptional termination cassette we showed to be effective mutagens with the PX and RP2 vectors. Given that the major functional change to RP8 lies in its 3’ exon trap, we expect its mutagenicity to be similar to RP2, but we don’t have the data to confirm that it is the same at this time. We therefore have explicitly acknowledged in subsection “Gene-break transposon system as a next generation mutagenesis system” that we have not performed the analyses of RP8 mutagenicity and kept the RP8 vector description with this caveat.

4) In this manuscript, they described the nature of the traps mainly by based on the gene information (in relation to human disease genes, gene ontology (PANTHER), and subcellular localization). The power of this system is visualization of subcellular localization of the trapped protein. The expression patterns can be visualized without the protein trap mechanism. They need to show the data for subcellular localization of a certain number of the trapped proteins (at least more than 10?). They should show co-localization of the fusion protein and endogenous protein if the antibody is available.

We certainly appreciate the concern about subcellular localization data gained through GBT protein trapping. In our previous work, the subcellular localizations have, in most cases, been consistent with what is expected. These observations lead us to perform the new analyses based upon PANTHER and Human Protein Atlas instead using a high resolution imaging approach. We realize that we did not explicitly state this or cite these previous publications in this section. In the revision, we have added citations to our previous work and our rational for these analyses.

Similarly, we agree that showing some new protein trapping expression data, especially some that we can relate to our PANTHER and/or Human Protein Atlas analyses, is useful for readers. We have therefore included confocal images of the GBT-confirmed lines GBT0235 (*lrpprc*) and GBT0348 (*ryr1b*) to highlight distinctive subcellular localization patterns at the top of the revised Figure 6. The human ortholog of *lrpprc* maps to mitochondria in Human Protein Atlas, but fails to map to a PANTHER class. The human ortholog of *ryr1b* maps to the cytosol, Golgi, and vesicles in Human Protein Atlas and is classified as a transporter in PANTHER analysis. We have also added additional confocal images that show distinct subcellular localizations in GBT-candidate lines and GBT lines to further demonstrate protein trapping in these lines in Figure 6—figure supplement 1.

5) ryr1b gene and mutants had been extensively reported by Hirata et al., (2007) and by themselves (Clark et al., 2011). What is new here is calcium imaging of muscle activities in the ryr1b mutant. However, the same result was obtained and described in Hirata et al., (2007) by electrophysiology. Thus, the novelty is poor. They should describe analysis of another gene which was not described previously.

The inadvertent omission to the prior and excellent Hirata paper has been corrected. Hirata et al., 2007 did a calcium imaging approach to assess the muscle specific phenotype but only quantified peak amplitude.(Hirata et al., 2007) We additionally quantified kinetic parameters in our calcium imaging data. Further, we have re-emphasized these experiments primarily serving as validation study rather than bringing substantial novel data to the table.

6) They identified 12 homozygous lethal mutants, but I do not see any data for this. The authors should show the data. 5 mutants they mentioned in the text were already described in the previous manuscript.

We have developed a table (Supplementary file 2 in the revision) characterizing all published mutants generated with the GBT system to date.

7) In Table 3, they say they identified 12 new previously-unannotated genes. What are those genes? and how they were trapped by the vector? More data should be shown.

We appreciate the interest in the previously unannotated transcripts. We have spent additional time analyzing these loci using additional publicly available resources, our own in-house RNA-Seq datasets and GRCz10. We focused on WT embryos at 4 different developmental stages, 36 hpf, 48 hpf and 4 dpf (Data source: ftp://ftp.ensembl.org/pub/data_files/danio_rerio/ across). The detail of this analysis is described in the revised method section. At the end of this additional work, we were able to resolved 7 lines (GBT0148, GBT0264, GBT0724, GBT1024, GBT1100 and GBT1168) into new cloned loci with likely transcripts (in the revised Supplementary file 1). In total, we resolved 7 out of these 10 unannotated GBT lines (in the revision) and moved this into our main catalog. Since we have informed genomic location of all 204 integration loci in the revised Supplementary file 1, we removed the previous Table 3 with the redundancy. We have also removed the individual section discussing the previous Table 3. Instead, we have added the following text:

Subsection “Molecular cloning of a subset of GBT lines highlights the genetic diversity of this protein trap collection”, “A small subset of these GBT-confirmed lines mapped to areas in the genome without annotated transcripts. Publicly available RNA-sequencing data revealed reads flanking a majority of these mapped integrations. Some of these reads contained evidence of splicing in the sense orientation of the mRFP reporter (Supplementary file 1)”.

Subsection “GBT protein trapping provides the basis for annotation of functionally diverse proteins and novel transcripts mapped on poorly assembled genomic regions” now reads, “We indeed found that 10 population of GBT integrations in the confirmed lines (with mRFP expression) failed to map to any predicted gene. However, RNA sequencing reads in public datasets identified potential unannotated coding sequences aligned with these GBT integration loci (Supplementary file 1). While 5' and 3' RACE are necessary to confirm the mRNA fusion products, these unannotated coding sequences represent the possibility to annotate novel, protein-coding transcripts in these GBT lines. Therefore, GBT protein trapping can find, illuminate expression, and elucidate in vivo functions of novel genes and/or gene variants in poorly annotated regions of reference genomes”.